

# Strategies for improving the communication of satellite-derived InSAR ground displacements

C. Scott Watson[1], John R. Elliott[1], Susanna K. Ebmeier[1], Juliet Biggs[2], Fabien Albino[3], Sarah K. Brown[4], Helen Burns[5], Andrew Hooper[1], Milan Lazecky[1], Yasser Maghsoudi[1], Richard Rigby[5], Tim J. Wright[1],

[1]COMET, School of Earth and Environment, University of Leeds, Leeds, LS2 9JT, UK
[2]COMET, School of Earth Sciences, University of Bristol, Bristol, UK, BS8 1RJ, UK
[3]ISTerre, Grenoble, France
[4]School of Earth Sciences, University of Bristol, Queens Road, Bristol, BS8 1RJ, UK
[5]CEMAC, School of Earth and Environment, University of Leeds, Leeds, LS2 9JT, UK

Correspondence to: C. Scott Watson (c.s.watson@leeds.ac.uk)

**Abstract.**

Satellite-based earth observation sensors are increasingly able to monitor geophysical signals related to natural hazards, and many groups are working on rapid data acquisition, processing, and dissemination to data users with a wide range of expertise and goals. A particular challenge in the meaningful dissemination of Interferometric Synthetic Aperture Radar (InSAR) data

to non-expert users is its unique differential data structure and sometimes low signal to noise ratio. In this study, we evaluate the online dissemination of ground deformation measurements from InSAR through Twitter, alongside the provision of open access InSAR data from the Centre for Observation and Modelling of Earthquakes, Volcanoes and Tectonics (COMET) Looking Into Continents from Space with Synthetic Aperture Radar (LiCSAR) processing system. Our aim is to evaluate (1) who interacts with disseminated InSAR data, (2) how the data are used and (3) to discuss strategies for meaningful

communication and dissemination of open InSAR data. We found that InSAR Twitter activity was primarily associated with natural hazard response, specifically following earthquakes and volcanic activity, where InSAR measurements of ground deformation were disseminated, often using wrapped and unwrapped interferograms. For earthquake events, Sentinel-1 data were acquired, processed, and tweeted within 4.7±2.8 days (shortest was one day). Open access Sentinel-1 data dominated the InSAR tweets and were applied to volcanic and earthquake events in the most engaged with (retweeted) content. Open access

InSAR data provided by LiCSAR was widely accessed, including automatically processed and tweeted interferograms and interactive event pages revealing ground deformation following earthquake events. The further work required to integrate dissemination of InSAR data into longer-term disaster risk reduction strategies is highly specific, both to hazard-type, international community of practice, and local political setting and civil protection mandates. Notably, communication of uncertainties and processing methodologies are still lacking. We conclude by outlining the future direction of COMET

LiCSAR products to maximise their useability.



## 1 Introduction

### 1.1 Capabilities and communication of SAR and InSAR data

There is a contrast between the intuitive accessibility and processing requirements of 'visual' optical satellite imagery compared to the active microwave sensing of synthetic aperture radar (SAR) (Table 1). Similarly, limited availability of open access to SAR satellites has slowed the operational and scientific uptake until the Copernicus Sentinel-1 programme started in 2014. This availability of open access Sentinel-1 data over the last ~8 years, in addition to other emerging SAR satellites, creates an opportunity for making SAR and InSAR data accessible and useable to non-experts.

Optical satellite imagery from the Landsat Multispectral Scanner System (launched 1972) preceded SAR (European Remote-Sensing Satellite - launched 1991) by nearly two decades and opened the door for space-borne scientific research. Prior to this, United States reconnaissance satellites operated from the 1960s; however, these optical images were only declassified in the last few decades (Fowler, 2013). Optical imagery is intuitive to interpret by the human eye and is therefore interpretable by non-specialists (Voigt et al., 2007), even considering non-visible spectral information such as near- or short-wave infrared, which can reveal non-visible characteristics such as the presence of chlorophyll and therefore the health of vegetation (Figure 1a) (Pettorelli et al., 2005). By comparison, SAR data, representing transmitted radio wave returns from a side-looking antenna, produce an all-weather view of the earth characterising the scattering properties of the reflecting surface or object (Rosen et al., 2000) but are less readily interpretable. The image brightness or intensity is influenced by the reflecting surface characteristics including texture, shape, water content, and satellite viewing angle (Gens and Van Genderen, 1996) (Figure 1b). The viewing angle can also cause shadowing and layover issues for steep terrain or for tall objects (Hagberg and Ulander, 1993). The SAR acquisition mode, wavelength, and polarisation also add to the complexity of data that is acquired due to variable radar penetration depths and interaction with surface features. The recent expansion in commercial SAR constellations (e.g. ICEYE and Capella), which often focus on providing derived products and results rather than raw data, also influences the future interpretability requirements of SAR data.

Interferometric Synthetic Aperture Radar (InSAR) uses pairs of radar images to measure radar wave phase differences, which corresponds to ground displacement after applying geometric, topographic and atmospheric corrections (Simons and Rosen, 2007). InSAR measurements of ground displacements are used to study natural hazards and processes such as earthquakes (Wright et al., 2004; Massonnet et al., 1993), landslides (Lauknes et al., 2010; Calò et al., 2012), volcanoes (Hooper et al., 2004; Ebmeier et al., 2018), and ice flow (Goldstein et al., 1993), in addition or anthropogenic signals such as ground water extraction and subsidence (Colesanti et al., 2003). For large and complex deformation, displacements derived from optical satellite imagery are a valuable alternative, particularly using daily revisit constellations such as Planet cubesats (Kääb et al., 2017). Over the last decade, new commercial radar satellites and open access Sentinel-1 data have transformed InSAR into a routine analytical and monitoring technique (Biggs and Wright, 2020).




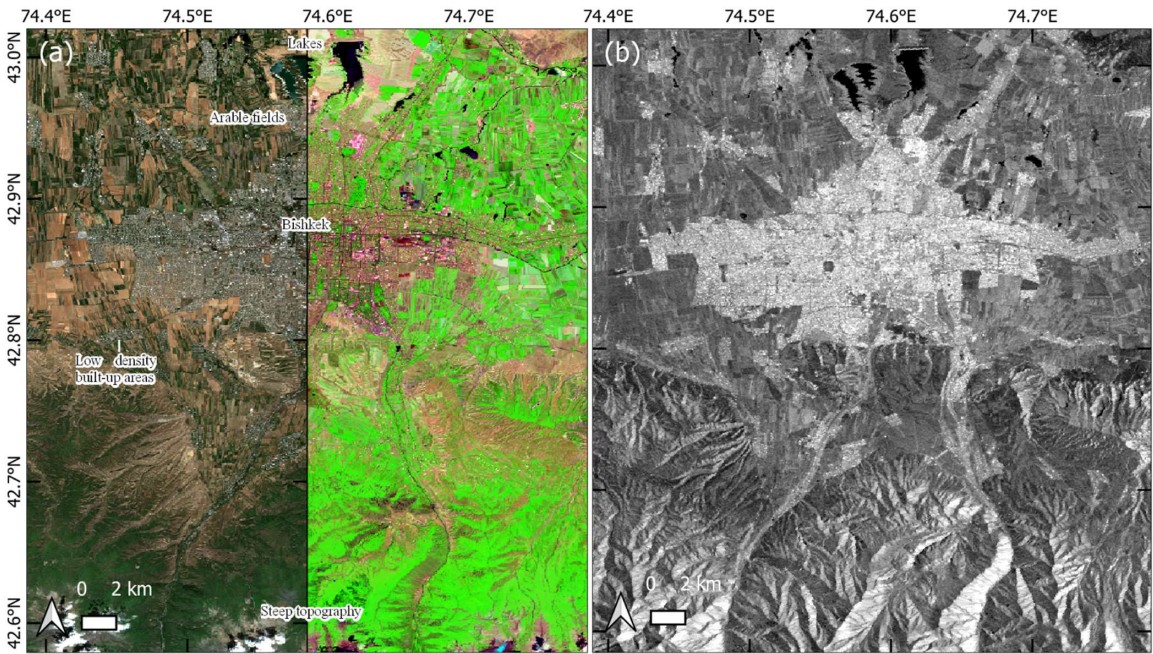

**Figure 1: Comparing optical and SAR data (a) Example of a 10 m resolution Sentinel-2 optical image (27th June 2021) over the city of Bishkek and mountains to the south. Left panel shows a true-colour (red, green, and blue bands) image and the right panel shows a false colour composite of (shortwave infrared, near infrared, and blue bands). (b) Sentinel-1 radar multi-looked intensity image**
**(26th June 2021) from the Looking Into Continents from Space with Synthetic Aperture Radar (LiCSAR) processing system (Lazecký et al., 2020) covering the same area of Bishkek in (a) at 50 m resolution.**

Common products generated using InSAR include wrapped and unwrapped interferograms, coherence maps, and displacement timeseries (Figure 2, Table 1). Wrapped interferograms show colour cycle fringes representing λ (wavelength)/2, equivalent to 2.8 cm of motion per cycle for Sentinel-1 towards or award from the satellite, which are unwrapped to show a line-of-sight

displacement in an unwrapped interferogram (Figure 2a). Coherence is low if the scattering properties of the reflecting surface change between SAR acquisitions, which can inhibit displacement measurements (e.g. close to a fault rupture following an earthquake but more commonly associated with changes in vegetation or snow cover). However, changes in coherence also reveal insights into other Earth surface processes such as glacier movement (Atwood et al., 2010) (Figure 2c), volcanic eruption (Dualeh et al., 2021) and landsliding (Burrows et al., 2019), or urban damage following disaster events (Pilger et al., 2021).

Routine or on-demand production of InSAR products is available through several platforms including LiCSAR (Lazecký et al., 2020), the Alaska Satellite Facility (Kennedy et al., 2021; Meyer et al., 2017), and the GeoHazards Exploitation Platform (Galve et al., 2017). The LiCSAR processing system is detailed by Lazecký et al. (2020). Briefly, Sentinel-1 SAR data are



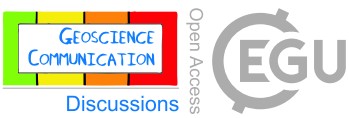

automatically processed over active tectonic and volcanic regions to primarily produce coherence maps, wrapped and unwrapped interferograms at 0.001° resolution (111 m at the equator), which can be used to derive displacement time series

using LiCSBAS software (Morishita et al., 2020). Open access datasets and online analytical tools and geographic information services can broaden the user base of satellite data by reducing the requirement for complex processing expertise. This is important when developing products and systems that integrate earth observation data, for example in-country monitoring capabilities for natural hazards to reduce disaster risk, in addition to using the data for disaster response (Joyce et al., 2009; Wang et al., 2012; Poblet et al., 2014).

**Table 1: Interpretability of common InSAR products**

| Product | Summary | Interpretation challenges |
|---|---|---|
| Coherence | A measure of stability of the scattering surface through time. | • Multiple potential sources of coherence change including vegetation, snow, water, ground deformation.<br>• The magnitude of coherence change is a function of time.<br>• Dependent on relationship between slope and satellite geometry.<br>• Varies markedly with SAR wavelength. |
| Wrapped interferogram | Ambiguous measure of apparent surface deformation in cycles of phase (fringes). | • Observed signal could be dominated by topographic or other atmospheric effects such as from the ionosphere and be misinterpreted as a deformation signal.<br>• Observed signal is relative to the satellite along a single look direction, without known 3D components. Therefore, determining the direction of the deformation is non-intuitive and non-unique.<br>• Interferograms may contain seasonal signals (e.g. related to vegetation), which are overprinted on any observed deformation signal. |
| Unwrapped interferogram | Cumulative measure of apparent surface deformation in the satellite line of sight. | • Observed signal could be dominated by topographic or other atmospheric effects such as from the ionosphere and be misinterpreted as a deformation signal.<br>• Observed signal is relative to the satellite along a single look direction, without known 3D components. Therefore, determining the direction of the deformation is non-intuitive and non-unique.<br>• Filtering and unwrapping methods influence the observed signal.<br>• Interferograms may contain seasonal signals (e.g. related to vegetation), which are overprinted on any observed deformation signal. |
| Time series | Temporal deformation change derived from a network of interferograms. | • Measurements are relative to a reference area, which is assumed to contain no deformation (unless tied to another data source such as GNSS).<br>• Long term biases may be present over some land covers in particular those containing vegetation (phase bias).<br>• Time series may contain seasonal trends (e.g. related to vegetation or atmospheric contribution), which are overprinted on any observed deformation signal.<br>• The atmosphere (water vapour) can introduce errors in the timeseries. Corrections using a variety of methods including weather models and GPS stations can remove the atmospheric effects; |



however, these techniques may not work or increase errors where the atmosphere is highly variable, for example over steep and high relief topography.

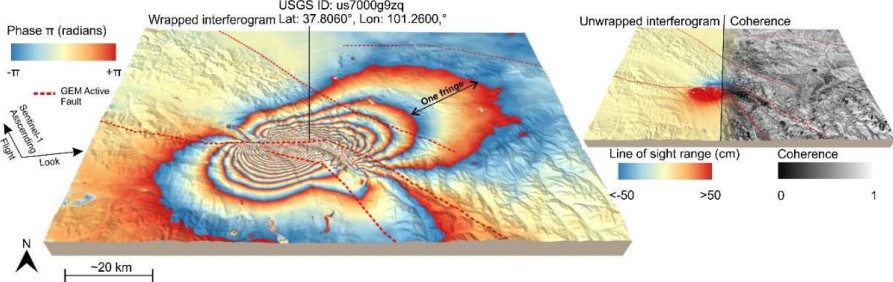

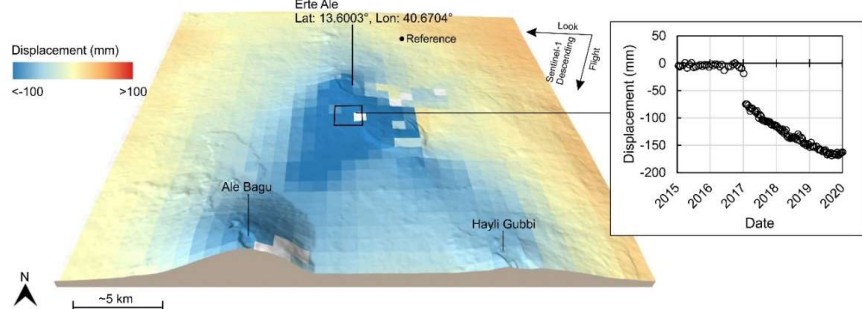

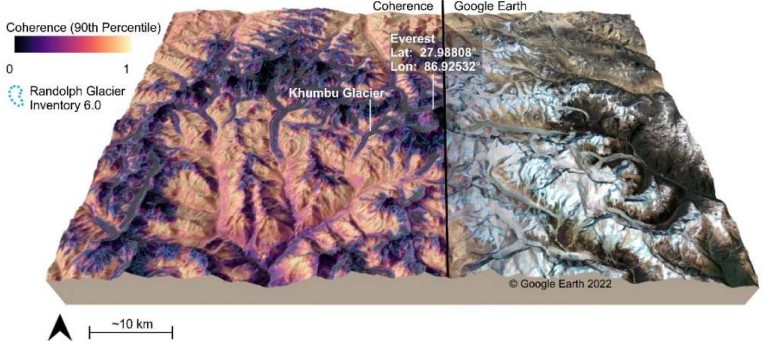

**Figure 2: Example applications of InSAR products from the LiCSAR processing system (Lazecký et al., 2020). (a) Wrapped and unwrapped interferograms and coherence data for the M6.6 earthquake in north-central China (7ᵗʰ January 2022). (b) Displacement time series for Erte Ale Volcano capturing the 2017 eruption (https://comet.nerc.ac.uk/comet-volcano-portal/). Higher resolution**





displacement timeseries were presented by Moore et al. (2019). (c) Coherence and optical satellite data of glaciers in the Everest Region.

### 1.2 Natural hazards communications

Both optical and radar satellite sensors, often in combination, are used to monitor and respond to natural hazards owing to their
large spatial coverage, increasingly short revisit periods, and the diverse scope of analytical capabilities (Elliott et al., 2016; Kirschbaum et al., 2019; Joyce et al., 2009; Tronin, 2010). Data are acquired and delivered in response to activations of the International Charter on Space and Major Disasters (Voigt et al., 2007), or through ongoing monitoring from satellites such as the Landsat and Sentinel series. For emergency response, data are analysed to produce mapping products (e.g. of building damage, floods, and fires), which are communicated to relevant in-country organisations to support response coordination,
such as through the Copernicus Emergency Management Service (Directorate Space, Security and Migration, European Commission Joint Research Centre (EC JRC), 2020). Satellite data that are usually restricted or expensive are also provided for research and hazard response through international initiatives such as the CEOS Working Group on Disasters Pilots and Demonstrators (Pritchard et al., 2018; Kirschbaum et al., 2017). Recently, social media sites have been used by scientists and citizen scientists to investigate disasters in real-time using satellite data and eyewitness multimedia and accounts to
collaboratively build knowledge in an open forum (Lacassin et al., 2020; Shugar et al., 2021; Ruan et al., 2022; Hicks, 2019; Earle et al., 2011). Twitter in particular has emerged as a platform used by scientists to create, discuss, and share research outputs (Van Noorden, 2014; Bruneau et al., 2021). National agencies such as the United States Geological Survey or volcano monitoring observatories also use dedicated social media accounts to provide updates on a range of natural hazards. These Twitter communications are open, therefore allowing direct public interaction, and are also increasingly directly embedded in
news media reporting (Oschatz et al., 2021; Broersma and Graham, 2013). This highlights the important ethical considerations regarding the release of data, with or without interpretation or communication of uncertainties, by users with differing expertise.

Effective communication and knowledge co-production between scientists and citizens is essential to creating effective risk communication and disaster risk reduction strategies (Hicks and Barclay, 2018); however, it requires negotiating complex
professional conduct considerations including: leadership, communications, cultural differences, community requirements and expectations, access to information, and misinformation (Newhall, 1999). Provision of open-source satellite data and analysis tools offers one mechanism to break down geographical and social barriers to knowledge co-production (Turner et al., 2015). However, the ethics of scientific conduct and data dissemination are complex, particularly during extended disaster events such as volcanic crises, for example as described by Newhall et al. (1999). Open access data means multiple overlapping
processing efforts are possible and have the potential to cause confusion if disseminated independently. Some examples of considerations for individuals or teams that remote to the event, include: (1) when and what data to release and with what level of interpretation, considering interpretation differences and underlying uncertainties; (2) considering whether releasing data

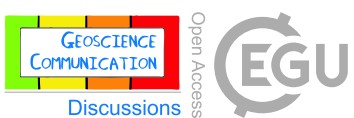

will distract from official advice, or undermine local teams; (3) whether releasing the data can contribute to public safety, even if results are only preliminary.

### 1.3 Study aim: LiCSAR and InSAR communication

In this study, we aimed to evaluate how InSAR data are communicated and engaged with through Twitter and the automatic InSAR processing system LiCSAR. Specifically, we address what we can learn from Twitter communication to improve the utility of this processing system. Through COMET, open-access LiCSAR data supports research into tectonic, volcanic, and other earth processes, in addition to responding to natural hazards and aiming to support the integration of earth observation data into disaster risk reduction strategies.

Noting the less intuitive visual outputs and applications of InSAR to non-specialists (Table 1), we explore whether communications are predominantly in-reach (to other academic scientists) or outreach (e.g. to industry experts, the media, and decision-makers), and the types of data that are communicated. We incorporate user feedback from LiCSAR to identify how the current provision of data products aligns with those communicated using Twitter. We also consider engagement with the LiCSAR Earthquake InSAR Data Provider (EIDP) that automatically processes and disseminates InSAR data for major earthquakes. We then identify future directions to improve the communication and application of InSAR products to natural hazard disaster risk reduction.

### 2 Data and methodology

### 2.1 InSAR tweets

We used the Twitter application programming interface (API) and Rtweet package v0.7.0 (Kearney, 2019; R Core Team, 2021) to retrieve public tweets (excluding retweets) 18th August 2020 to 9th February 2022 mentioning "InSAR". Twitter data are increasingly used in scientific analysis (Ruan et al., 2022; Lacassin et al., 2020; Acar and Muraki, 2011; Sinnenberg et al., 2016). We processed tweet text to remove stop words, mentions, uniform resource locators, emojis, numbers, and punctuation (e.g. Ruan et al., 2022). Finally, we removed a list of other words, tweets, and user IDs that were not relevant to our study, including "InSAR" tweets relating to the International Society for Autism Research (Supplementary information). We used word counts from the remaining tweet text to identify the most commonly occurring words or hashtags in the InSAR tweets to identify the context of the InSAR tweets.

To identify the most common and most popular InSAR Twitter communications, we examined the user networks for the top ten InSAR tweeters, (1) ranked by number of tweets, and (2) the tweeters of the top ten most retweeted tweets. For each of the top ten user IDs (a static ID number which unlike the Twitter user handle, cannot be changed) in (1) and (2), we used Rtweet to find the network of users that were followed by each account, therefore unravelling active components of the Twitter InSAR

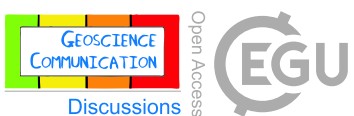

network. User connections were summed to find commonly connected users within the InSAR network. For example, a user ID followed by five of the top ten tweeters would feature a count of five. We also extracted all user IDs that followed the top ten accounts in (1) and (2). These followers represent the users that would receive tweets on their timeline from the top ten InSAR tweeters in (1) and (2). The profile description of these users were classified using key word search strings from Côté et al. (2018) into 'scientist' and 'non-scientific' groups using the R package 'stringr' (Wickham, 2017) to determine different applications of InSAR and whether InSAR communications were predominantly amongst scientists or outreach. The keyword strings we used are shown in the Supplementary information. Examples of the 'scientist' class included: 'professor', 'phd', 'post doc', 'doctoral', and 'research scientist' and could be within academia (universities) or applied industry applications. Profiles that did not match the scientist classification were assigned 'non-scientist'. 'Non-scientists' are expected to include decision-makers such as government agencies; media and journalists; outreach organisations such as museums and science educators, and other members of the public (Côté et al., 2018). These classifications were used to disaggregate different users of InSAR data; however, we acknowledge the distinction between 'scientist' and 'non-scientist' is ambiguous in some cases. We validated the classification by manually inspecting a random sample of 500 users from each of the scientist and non-scientific groups. We did not disaggregate the non-scientific group into additional categories such as 'media' or 'decision makers' due to the high misclassification rates observed by Côté et al. (2018).

### 2.1.1 Ethical considerations

Ethical considerations exists despite the public nature of Twitter data and prior usage consent being given though acceptance of Twitter's terms of service (Sinnenberg et al., 2016; Gold, 2020). In our study, we collected Tweets using the Twitter API, and report anonymised and aggregated statistics, other than reporting on the top tweeters of InSAR data, where we believe reporting a subset of Twitter usernames does not create a risk of harm, provides recognition to active members of the Twitter InSAR community, and adds context (e.g. Lacassin et al., 2020).

### 2.2 Website Analytics

To evaluate how open access InSAR data are used and by whom, we used statistics from the COMET LiCSAR project, combined with our analysis of InSAR communications using Twitter.

Website analytics for the LiCSAR portal were obtained from Google Analytics from January 2017 to December 2019, and WordPress.com starting in 2020. Both platforms use a page 'views' metric, which represents instances of a user viewing the portal web page; however, the two analytics are likely not directly comparable. User feedback on the LiCSAR Portal was collected using a Google Forms survey containing 15 questions (Supplementary information), which were designed to collect information on who was using the portal, their usage and experience with LiCSAR data, and suggestions for future developments. Responses were provided by visitors to the LiCSAR portal that clicked the 'Give User Feedback' link on the portal homepage. Responses were also solicited by COMET members through peer networks and existing partnerships.



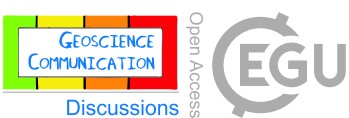

COMET membership was asked as a survey question to distinguish these responses. In this study, we used anonymised responses obtained from the initiation of the survey in February 2020 to February 2022 (n=80); however, a link to the survey remains on the LiCSAR Portal for ongoing feedback.

### 2.3 LiCSAR Earthquake InSAR Data Provider (EIDP)

LiCSAR data are produced following specific earthquake events and consolidated through an 'Earthquakes' tab on the LiCSAR
Portal. First, earthquake events that meet a threshold magnitude and depth, trigger the activation of LiCSAR frames for data processing (Lazecký et al., 2020). These events are plotted on an interactive JavaScript *Leaflet* map along with key metadata from the United States Geological Survey (USGS). The map interfaces with a data table that allows users to search and download data for each event from an online event page. These event pages are automatically generated for each earthquake event and are regularly updated in the following weeks to include new coseismic and post-seismic data. Details of the
earthquake events with a link to the event page are automatically tweeted by @COMET_database, in addition to the first processed interferogram overlapping the earthquake epicentre. We used page views logged using Google Analytics to evaluate user engagement with the EIDP system and we report statistics for the same reporting period as the Twitter data (18th August 2020 to 9th February 2022).

### 2.4 COMET Volcano Deformation Portal

We take a different approach to making LiCSAR data accessible for volcanic hazards, based on tailored, online tools designed for volcano observatory scientists. While InSAR measurements have the potential to capture displacements significant for understanding and even forecasting volcanic hazards, there are specific obstacles to the exploitation of satellite radar for volcanology. In 2017, we sent a short questionnaire to volcano observatories in developing countries and overseas territories,
designed to understand their current operational use of remote sensing, the issues they face, and what could be beneficial to them. Ten responses were received to the questionnaires, from Guatemala (x 2), Costa Rica (x 2), Ethiopia (x 2), Democratic Republic of Congo, Trinidad and Tobago (SRC, with responsibility for the English-speaking Caribbean), Montserrat and Indonesia. Interestingly, where more than one response was gathered from the same institution, answers sometimes differed. These differences did not arise in the usefulness of tools, but in the understanding of what was currently used and even the
number of volcanoes.

The key barriers to the use of InSAR data highlighted from these responses is summarised in Table 2. All those who responded agreed that they would find ready-processed and analysed data to be beneficial to their operations, improving their monitoring capacity. They would use such data to inform their deployment of ground-based monitoring instrumentation and to alert the
necessary authorities of unrest. The automated tools must be accessible without the need for high-speed internet or advanced computing facilities. Useful outputs would be short reports explaining the main points of the data analysis and findings, the metadata, and small JPEG images. These data need to be delivered in a timely manner and support would be required to ensure





proper understanding. Though ready-analysed data is desirable, the responders also stated that they would like training to accompany these resources, so they have a good understanding of the outputs and the processes leading to these outputs.


**Table 2: Barriers to uptake of satellite data identified by volcano observatories. Information is summarised from the responses from a range of official development assistance countries to a 2017 survey.**

| Issue | Explanation | Solutions |
|---|---|---|
| Awareness | Some observatories are not fully aware of the types of remote sensing available or where to access the data. | Collaboration. Capacity building. |
| Reliability | The internet and particularly social media means the world is more connected than ever before. Data are frequently posted online, especially during eruption crises, but observatories must ensure they are from a reliable source. | Automated processing. External verification. |
| Timeliness | In rapidly-developing unrest or eruption situations, timely access to data is crucial. Some data cannot be accessed in real-time or near real-time. The download and processing time of other data renders this inappropriate. | Automated processing. |
| Background knowledge | To understand the unrest at a volcano some knowledge of background activity is required. | Automated processing. |
| Ground truth | Ground-truthing can be required to verify what is seen through remote-sensing. This requires on-the-ground expertise, human and economic resources. | Collaboration. |
| Data cost | Some remotely-sensed data and software for analysis is prohibitively expensive | Collaboration. Data sharing. |
| Human resources | Training in specialist areas such as satellite data retrieval is costly. There may be no redundancy within a country, with one expert and no back-up. | Capacity building. |
| Computing resources | Processing data can require significant computing resources, including (often costly) specialist software, fast computers with powerful processors, large amounts of data storage space. | Automated processing. |
| Power supply and internet | Power can be unreliable in many countries, and particularly at remote observatories. Internet access and speed may be insufficient to download large data files. | Automated processing & small files. |

To address these issues, we have designed a set of online tools (COMET Volcanic and Magmatic Deformation Portal

(https://comet.nerc.ac.uk/comet-volcano-portal/)) to allow observatory scientists to critically interrogate automatically generated Sentinel-1 interferograms and identify deformation around active volcanoes (Rigby et al., 2021). The volcano portal provides access to interferograms and time series (Lazecký et al., 2020; Morishita et al., 2020) for an initial subset of volcanoes where processing is most complete, with the aim of eventually including all subaerial Holocene volcanoes. The portal allows users to browse through images of interferograms using a 'slider' tool and to plot timeseries (including choice of reference



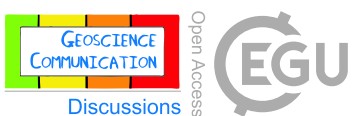

area) and profiles for comparison with topography interactively. Wrapped and unwrapped interferograms can be compared, as well as time series with and without GACOS atmospheric correction (Yu et al., 2018) and spatiotemporal filtering (Morishita et al., 2020). The aim is to provide users with tools that support interrogation of the data, with a particular focus on distinguishing between atmospheric features and volcanic deformation (e.g. Pinel et al., 2011; Ebmeier et al., 2018; Biggs et al., 2021). We present new Sentinel-1 InSAR results in the context of historical observations of volcano deformation (as

described in Ebmeier et al., (2018)) organised according to volcano numbers from the Smithsonian list of Holocene volcanoes (Global Volcanism Program., 2022). Machine Learning based tools (Biggs et al., 2022) that flag high magnitude displacements also provide users with a tool for quickly finding major deformation events in a time series. We deliberately do not provide authoritative interpretations of the interferograms, but provide tools for an engaged user to assess the automatically processed data for themselves, facilitating comparisons to both ground-based measurements and independent processing of the InSAR

data. The spatial resolution of images used for the on-the-fly time series analyses is low, being based on boxes centred on volcanoes (or sites of active magmatic deformation) with a modal length and width of 50 km (this would be sufficient to capture ~90% of historical magmatic and volcanic displacements, although in a few cases signals do exceed ~100s km in diameter (Ebmeier et al., 2018; Henderson and Pritchard, 2013). These tools are designed to allow on-the-fly analysis, and also allow the user to download either the time series results shown on screen (currently in csv format) or the original

interferograms. Our communication strategy for volcanic signals is in contrast to the more responsive role of the EIDP, and reflects the challenges of forecasting risk associated with volcanic hazards that evolve dynamically on temporal scales that can range from hours to years.

## 3 Results

### 3.1 InSAR Twitter

We collected a total of 16,621 tweets from 18th August 2020 to 9th February 2022 mentioning "InSAR", which was reduced to 5,349 after removing retweets, and down to 4,312 after cleaning the dataset (Section 2.1). These cleaned tweets originated from 1,935 unique Twitter users. We observed a mean of 8±7 tweets per day and spikes related to notable events including scientific conferences, news media reports, earthquakes, and popular Twitter threads (Figure 3). A data gap was present in the timeseries (29/10/2020–06/12/2020); however, the source was unknown and could be related to an issue with the collection of

tweets. Specific earthquake events related to increased activity included those in Croatia (USGS ID: us6000d3zh. Date: 2020-12-29), Greece (USGS ID: us7000df40, Date: 2021-03-03), China (USGS ID: us7000e54r, Date: 2021-05-21), and Haiti (USGS ID: us6000f65h, Date: 2021-08-14). Typically, these events included multiple other earthquake events (foreshocks and aftershocks) at each location. Other peaks in tweets were related to InSAR sessions at the American Geophysical Union (AGU) and European Geosciences Union (EGU) conferences in December 2020 and April 2021 respectively. News media reports

were associated with tweet peaks in June 2021 related to the upcoming Venus Emissivity, Radio Science, InSAR, Topography, and Spectroscopy (VERITAS) mission to map Venus, and to reports of subsidence in Mexico City. Finally, two peaks in tweet



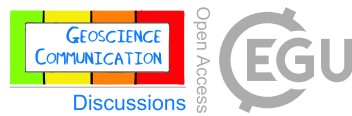

activity were related to specific Twitter threads written in Spanish on subsidence in Mexico City, and earthquake seasonality
in Mexico (Figure 3).

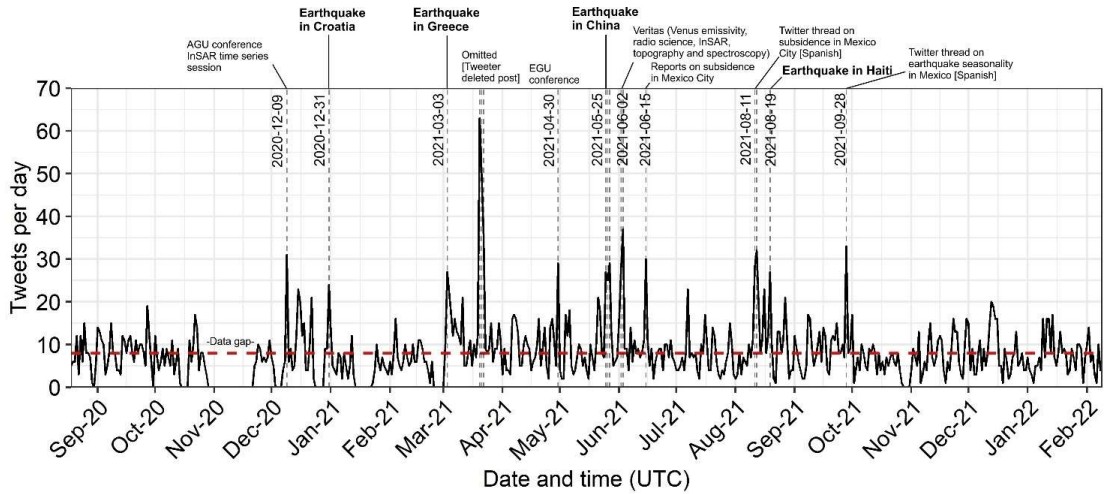


**Figure 3: Timeline of InSAR tweets. Dashed red line shows the mean of eight tweets per day (excluding data gap). Vertical lines and annotation are shown for events exceeding 22 tweets per day (two standard deviations from the mean). Dates shown for the earthquake events in bold represent the date of peak Twitter activity. The peak in March 2021 was omitted since the post and account were removed by the user.**


Word count analysis of the text of InSAR tweets revealed that mentions of 'data', including Sentinel-1 SAR, and applications
of InSAR, such as to natural hazards, occurred most frequently (Figure 4). 'Data' was the most frequently occurring word (n=
676) followed by 'Earthquake' (n= 312) and 'Deformation' (n= 269) (Figure 4a, d). The most popular hashtags used in InSAR
tweets were 'Sentinel1' (n=323), 'Earthquake' (n=259), and 'SAR' (n=143) (Figure 4b). The most popular words and hashtags

in the top 50 most retweeted InSAR tweets were 'Deformation' (n=11), 'Earthquake' (n=10), and 'Dike' (n=9) (Figure 4c, e).
Specific data sources identified in the tweet word analysis included 'Sentinel-(1)', 'VERITAS', 'NISAR' (NASA-ISRO-SAR**)**,
and 'ALOS' (Advanced Land Observing Satellite) (Figure 4, Table S1).


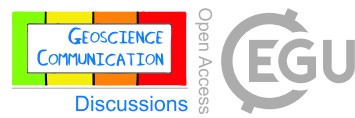

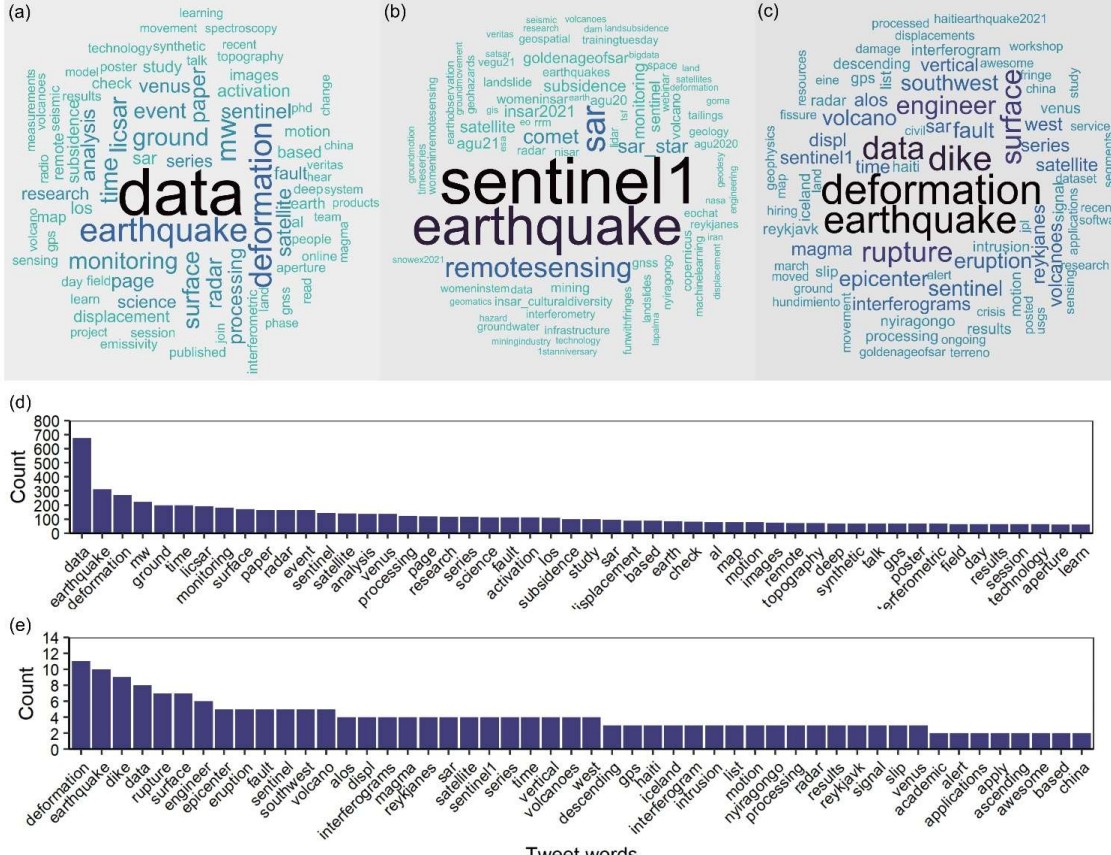

**Figure 4: (a) Word cloud showing the top 80 words occurring in the text of InSAR tweets. (b) Word cloud showing the top 80 hashtags occurring in the InSAR tweets. (c) Word cloud showing the combined top 80 words and hashtags occurring in the 50 most retweeted InSAR tweets. (d) Graphical count of the top 50 words in (a). (e) Graphical count of the top 50 words and hashtags in (c).**

We analysed the InSAR Twitter social network for (1) the top ten tweeters of InSAR content, and (2) the top ten most retweeted tweeters of InSAR content. Therefore, capturing (1) the greatest tweeters of InSAR related tweets, and (2) the tweeters of content with the highest level of engagement. The network plot (Figure 5a, b) circle markers show common connections and are coloured by their count (node_connections) where a user is followed by two or more of the top ten tweeters (Figure 5a), or a tweeter of a top ten most retweeted tweet (Figure 5b). The black lines with transparency represent other Twitter users followed by a labelled (black bold text) user. Two users were present in both groups: 'ericfielding' and 'mr_insar' (Figure 5a, b). The top ten tweeters followed 7,334 users of which 916 had two or more node connections (i.e. were followed by two or more of the top ten tweeters) (Figure 5a). By comparison, the top ten most retweeted tweeters followed 5,056 users, of which





687 had two or more node connections (Figure 5b). Users with high node connections (out of a potential maximum of ten), in some cases featured in both network plots (example users in blue text in Figure 5a, b) despite not being a top ten tweeter or retweeted tweeter themselves.

Figure 5: (a) User network showing who the top ten InSAR tweeters follow. Black lines with transparency represent individual users followed by each of the top ten tweeters. Circle markers show common connections and their count where a user is followed by two or more of the top ten tweeters. Users with greater than seven connections are labelled with their twitter handle. Note that 'ericfielding' is plotted as both a top 10 tweeter and a user with >7 connections. (b) Network plot as per (a) for the Twitter users that





**tweeted the top ten most retweeted tweets. (c) Paired word network from the user profile descriptions of scientists and (d) non-scientists. The count shows the number of times each pair of words occurred.**

The followers (n=31,680) of users in the top ten tweeter groups (black text in Figure 5a, b), were classified into scientist and non-scientific audiences to determine the in-reach and outreach status of the InSAR tweets. We excluded the space reporter 'thesheetztweetz' since their retweet was about the NASA VERITAS, which is a Venus mission we exclude in our study of Earth Science InSAR. We found that the networks were predominantly outreach, since 62% of the followers (n=19,778) were

non-scientists and 38% (n=11,902) were scientists. The profile description word network for the scientist group was more connected than non-scientists (Figure 5c, d). Commonly occurring paired words for scientist profiles that were indicative of user interests included 'machine learning' (n=113), 'natural hazards (n= 80), 'structural geology' (n=80), and 'active tectonics' (n=60) (Table S2). Non-scientist paired words were more diverse with lower occurrence counts, including 'civil engineer' and 'animal lover' (n=21), and 'gis specialist' (n=18). Commonalities between both communities included 'remote sensing', which

was most frequent in both lists and suggests the non-scientist group included a large community of satellite data users, likely including industry Earth Observation experts applying InSAR analyses.

Of the ten most retweeted InSAR tweets (Figure 6), two were plain text (a, e); one contained a video of volcanic deformation (b); one contained a photograph and description of subsidence (c); three contained wrapped interferograms of volcanic

deformation of Nyiragongo, Democratic Republic of the Congo (d), Iceland (g), and Galapagos (h); one contained new open-source SAR code (f); and two contained line of sight displacement measurements, one of volcanic deformation in Iceland (i) and one of an earthquake in Haiti (j). Four of the tweets presented Sentinel-1 data (d, g, h, i) and one presented ALOS-2 data (j). Notably, these tweets mostly featured data annotations and text explanations of the associated images and were retweeted a total of 1,487 times.





**(a)** Michael Sheetz
@thesheetztweetz **Retweeted: 510**

NASA announces two new missions to Venus:

1 - DAVINCI+ (Deep Atmosphere Venus Investigation of Noble gases, Chemistry, and Imaging Plus)

2 - VERITAS (Venus Emissivity, Radio Science, InSAR, Topography, and Spectroscopy)

https://twitter.com/thesheetztweetz/status/1400174073540493316

**(d)** Benoît Smets
@smetsbenoit **Retweeted: 92**

First Sentinel-1 interferogram over Nyiragongo and the Virunga region. Amazing deformation signal!
#Nyiragongo #OVG #GVO #Eruption #MasTer #InSAR

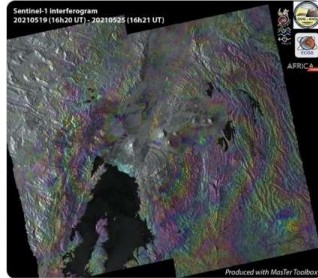

https://twitter.com/smetsbenoit/status/1397296480340353027

**(b)** Brendan Duffy
@structuregeo **[Video]** **Retweeted: 284**

Watch a volcano breathe #Etna #volcano #insar @NASAJPL photojournal.jpl.nasa.gov/archive /PIA132…

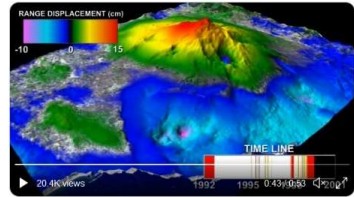

https://twitter.com/structuregeo/status/1306752548129267713

**(e)** Tesfaye T. Tessema
@TTemtime **Retweeted: 85**

Hello, my name is Tesfaye Temtime. I recently finished my PhD in Geophysics from the University of Bristol, UK. I am an enthusiast of detecting surface changes using satellite geodesy (i.e. InSAR and GNSS) and modelling. 🌍🛰️📡 #BlackInGeoscienceRollCall #BiGWeek2021

https://twitter.com/TTemtime/status/1435162379462062081

**(c)** Dario Solano
@Mr_InSAR **Retweeted: 172**

The #hundimiento of the land in #CDMX is due to the overexploitation of groundwater.
The same happens in many other cities in the country) #Guanajuato , #Guadalajara , #Toluca , #Puebla , etc.).
I open thread 🧵 of the sinking of the land in Mexico with little drawings made by me 👇
one/

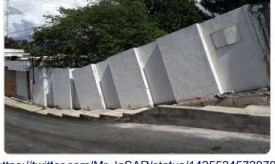

https://twitter.com/Mr_InSAR/status/1425524572070170628

**(f)** Franz J Meyer
@SARevangelist **Retweeted: 76**

🚨#SAR and #InSAR Twitter 🚨
Help us keep our curated list of awesome SAR software, libraries, and resources updated! Current list at
🔗: github.com/RadarCODE/awes…
Got new, awesome open-source SAR resources? Add them to the current list 🌵!
#OneStopSARShop #GoldenAgeOfSAR

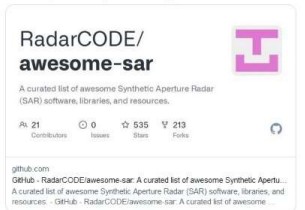

github.com
GitHub - RadarCODE/awesome-sar: A curated list of awesome Synthetic Apertu...
A curated list of awesome Synthetic Aperture Radar (SAR) software, libraries, and resources. - GitHub - RadarCODE/awesome-sar: A curated list of awesome

https://twitter.com/SARevangelist/status/1426704903108403201

**(g)** Sigurjón (Sjonni) Jónsson
@Sjonni_KAUST **Retweeted: 74**

Successive InSAR data of the Iceland crisis. The first includes the M5.7 earthquake, a few other offsets, and clear dike deformation, the second indicates dike migration to the southwest and shows the surface rupture of the M5 event last night (S1, desc., by Adriano) @ESA_EO

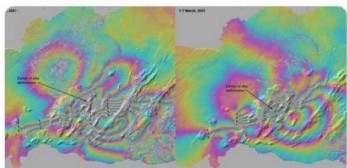

https://twitter.com/Sjonni_KAUST/status/1368589190850560000

**(j)** Dr. Eric J Fielding, PhD
@EricFielding **Retweeted: 60**

#Haitiearthquake2021 surface deformation measured by InSAR and SAR pixel offset tracking on JAXA ALOS2 images. Analyzed at NASA/JPL. Fault slip extends from epicenter about 60 km to west. Radar line-of-sight to satellite up and west. Both measure same direction in different ways.

**[Expanded tweet]**

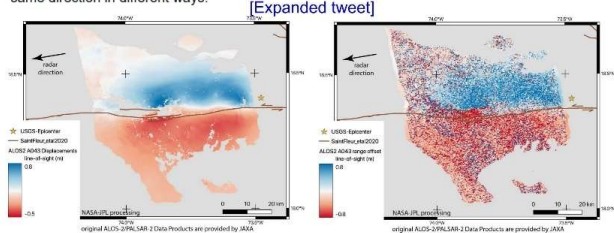

https://twitter.com/EricFielding/status/1428629519947034624

**(h)** TR Walter
@VOLCAPSE **Retweeted: 68**

I'd like sharing one of the most beautiful #InSAR pattern one can find at volcanoes. Thats from Wolf, Galápagos, showing intense deformation associated with the ongoing fissure eruption, captured by desc. Sentinel-1 in 5Jan22-17Jan22. Each colour cycle is approx 3 cm of motion.

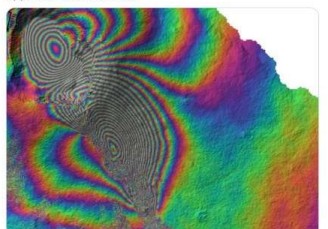

https://twitter.com/VOLCAPSE/status/1484616658345996292

**(l)** Kristín Jónsdóttir
@krjonsdottir **Retweeted: 66**

Latest InSAR also suggests a dike intrusion in the same location as this spring (24feb-19mar). Seismicity is still high and today a M4.7 at 15:03 occurred east of the crater (Stóri Hrútur). A new eruption has not started yet...
Volcano monitoring does not ask if it is holidays ;)

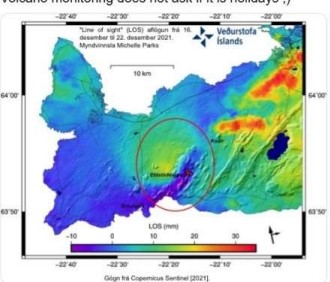

https://twitter.com/krjonsdottir/status/1474413422855008257





**Figure 6: The ten most retweeted InSAR tweets ordered (a)–(j) (1–10). The number of times each tweet was retweeted is shown adjacent to the Twitter username in red text. (Tweet citations: (a) Michael Sheetz [@thesheetztweetz] (2021); (b) Brendan Duffy [@structuregeo] (2020); (c) Darío Solano [@Mr_InSAR] (2021); (d) Benoît Smets [@smetsbenoit] (2021); (e) Tesfaye T. Tessema [@TTemtime] (2021); (f) Franz J Meyer [@SARevangelist] (2021); (g) Sigurjón (Sjonni) Jónsson [@Sjonni_KAUST] (2021); (h) TR Walter [@VOLCAPSE] (2022); (i) Kristín Jónsdóttir [@krjonsdottir] (2021); (j) Dr. Eric J Fielding PhD [@EricFielding] (2021)).**

### 3.2 LiCSAR Portal

The LiCSAR Portal (Figure 7a, b) disseminates automatically processed Sentinel-1 InSAR data for active tectonic and volcanic regions, thereby focussing on natural hazards, including a dedicated earthquake processing system known as EIDP. User feedback provided on LiCSAR and usage statistics for LiCSAR and EIDP were used to evaluate engagement with the system and how this related to InSAR communications over Twitter. Views to the LiCSAR portal homepage February 2021 to February 2022 (period of the user feedback survey) totalled 17,403, with an additional 3,538 views to the EIDP homepage (Table 3). Annual views increased by 4,833 (2020–2021) and averaged 1,148 views monthly (Figure 7c). Evaluation of the LiCSAR system was therefore expected to capture a range of users across the InSAR community.



**Figure 7: (a) LiCSAR data dissemination portal and (b) query system to evaluate the status of processing for each data frame using a range of filters (blue bars) to interactively filter the frames displayed on the map and data table. (c) Views of the LiCSAR portal homepage 2017–2021. Note that the 'views' analytics switched to WordPress from 2020 (Section 2.2).**




**Table 3: Select LiCSAR Portal page views February 2021 to February 2022**

| Page | Total views |
| --- | --- |
| LiCSAR Portal home[1] | 17,403 |
| Earthquakes (EIDP)[2] | 3,538 |
| Product details[3] | 2,002 |
| Velocities[4] | 1,936 |

1. https://comet.nerc.ac.uk/comet-lics-portal/
2. https://comet.nerc.ac.uk/comet-lics-portal-earthquake-event/
3. https://comet.nerc.ac.uk/comet-lics-portal-product-details
4. https://comet.nerc.ac.uk/comet-lics-portal-velocities

### 3.2.1 User feedback

User feedback on the LiCSAR Portal February 2020 to February 2022 revealed the profession of LiCSAR users and their views on current and future InSAR products (Figure 8). We report statistics using all responses (n=80); however, in some cases
not all questions were answered by respondents. Most users (83%, n=64) that provided feedback were not associated with COMET so were independent of the LiCSAR system. Respondents were from 29 countries and used LiCSAR data to study a total of 39 countries. Most respondents (77%, n=72) were academic (scientific) users. Geological/geophysics survey (7%) and public sector (6%) were the next highest groups (Figure 8a); however, the InSAR applications in these groups (scientific vs non-scientific) were not known. Users predominately used unwrapped and wrapped interferograms, and coherence products
(Figure 8b). We observed a broad spread in the number of interferograms used, with the majority using less than 100 interferograms and some using over 400 (Figure 8d). Similarly, most used fewer than five data frames, though some used over 250 (Figure 8e). Displacement time series, mean velocities, and subsidence/uplift rates were the most desired future products (Figure 8c). Overall ease of access to data through the portal averaged 3.8±1.1 on a scale of difficult (1) to easy (5) and most respondents identified a benefit to future integration of the data in a cloud-based platform such as Google Earth Engine, scoring
3.6±1.2 on a scale of no benefit (1) to strong benefit (5).





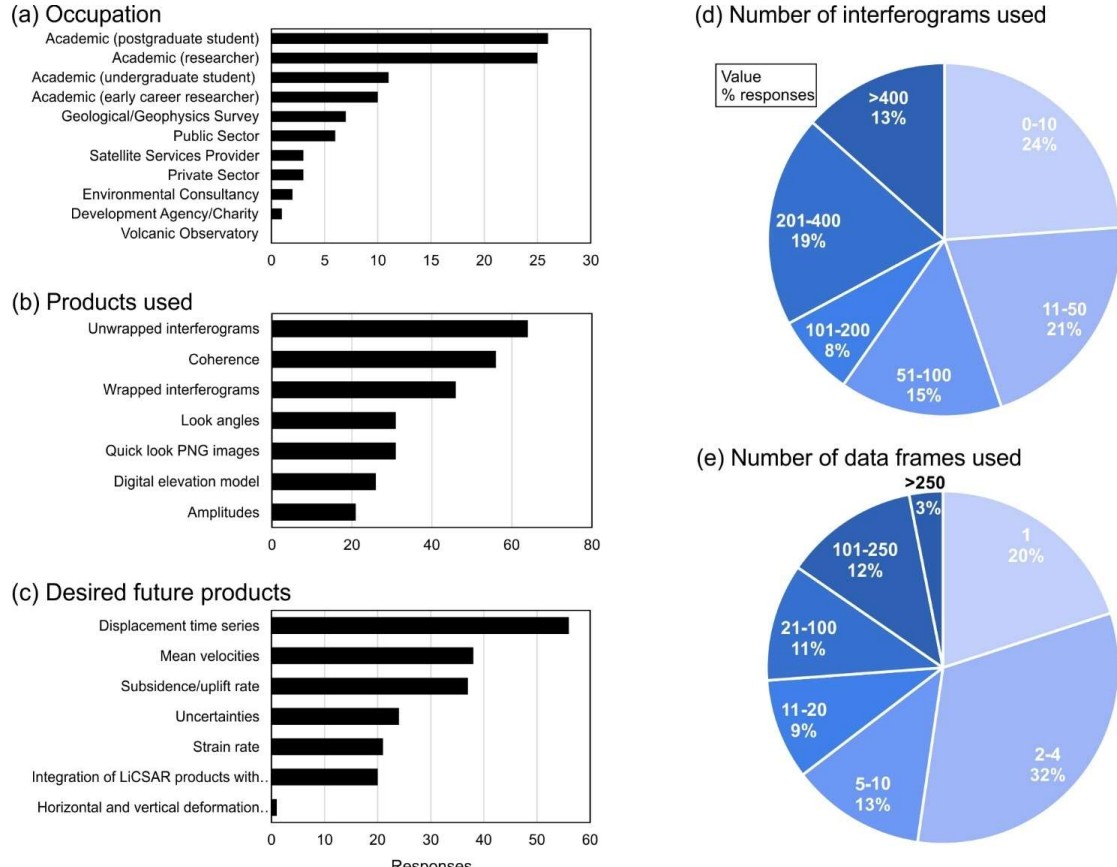

**Figure 8: Results from select questions in the LiCSAR data user feedback survey (February 2020 to February 2022, n=80). Additional survey results are discussed in the text.**

### 3.2.2 Earthquake InSAR Data Provider (EIDP)

The EIDP component of the LiCSAR Portal included 430 earthquake event activations between April 2019 and February 2022, which were plotted on the interactive event map (Figure 9a). The earthquake event pages and first wrapped interferogram generated over the epicentre (Figure 9b) were automatically tweeted by @COMET_database, which featured in the top ten InSAR tweeting accounts (Figure 5a) and accounts for the high instance of 'LiCSAR' word occurrence (n=190) in the InSAR tweets (Figure 4a, d). The five most retweeted EIDP interferograms all displayed a clear earthquake signal, shown by the colour fringe cycles and black arrow (Figure 10a). Three of these interferograms corresponded to earthquakes that were associated with peaks in tweet activity identified in Figure 3 (China 21/05/2021, Croatia 29/12/2020, and Greece 03/03/2021).



EIDP tweets coseismic interferograms regardless of the presence of a deformation signal and five examples with no earthquake signal are shown in Figure 10b, all of which were only retweeted once.

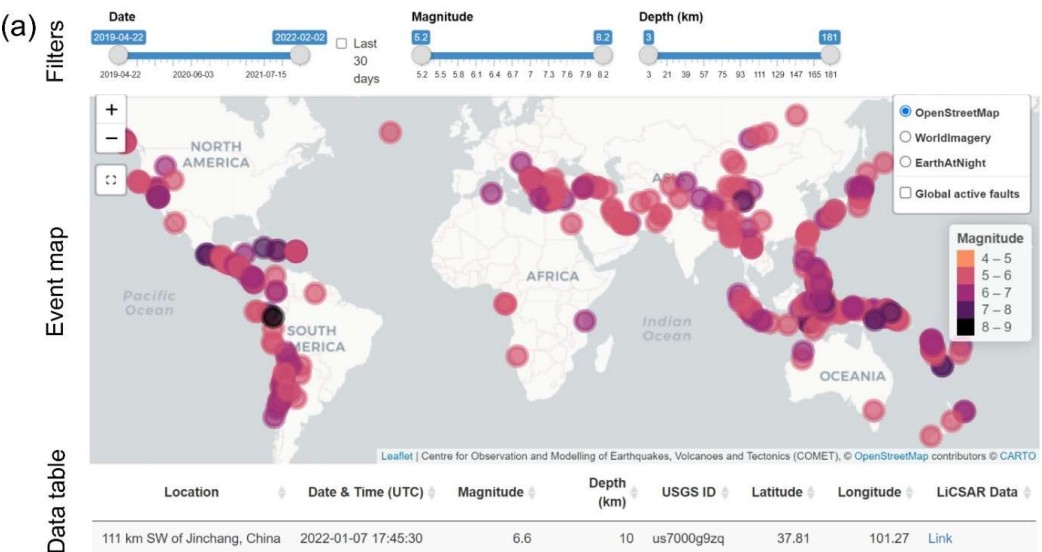

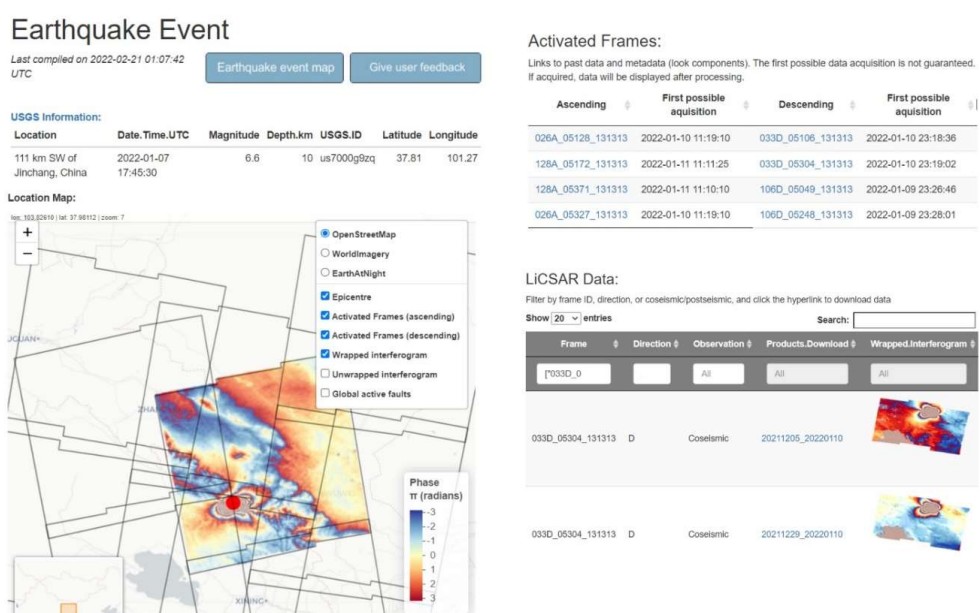


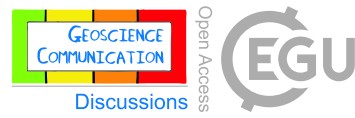

**Figure 9: Earthquake Event Data page and catalogue. (a) Earthquake parameters provided by USGS are shown in a table and interactive location map. (b-c) Event pages show USGS earthquake parameters and an interactive map showing activated frames, the first available interferogram data, and the option to overlay a fault map and change the basemap layers. Full resolution event pages are available at** https://comet.nerc.ac.uk/comet-lics-portal-earthquake-event/**.**


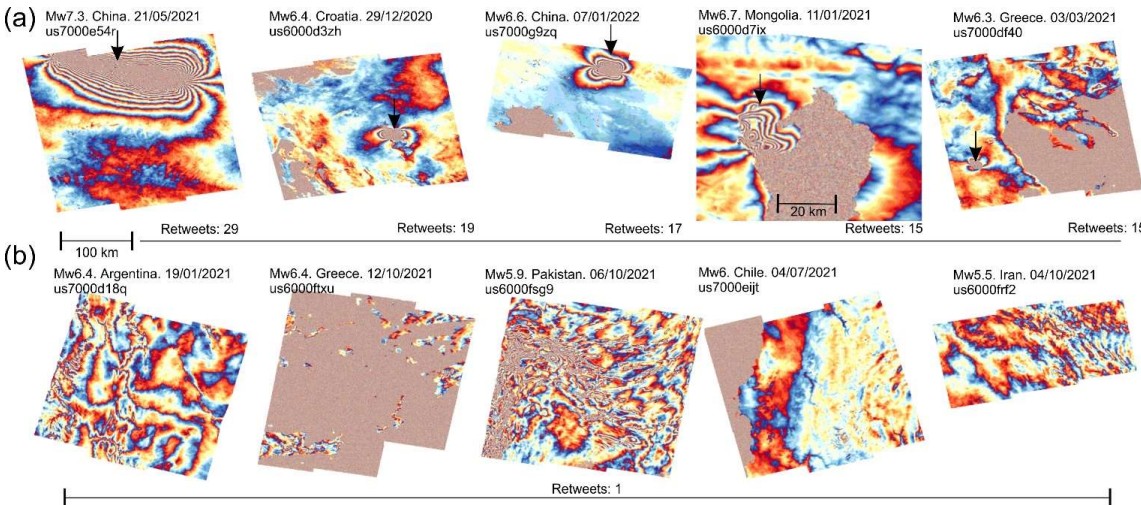

**Figure 10: Examples of wrapped interferograms tweeted for earthquake events (magnitude, date, and USGS ID are annotated) by @COMET_database, including the five most retweeted interferograms (a) with the earthquake signal annotated with the arrow, and five interferograms without an easily-identifiable earthquake signal (b). Note that the 100 km scale bar is applicable to all panels**
**except the Mongolia earthquake (us6000d7ix).**

We evaluated the engagement with EIDP using page views logged through Google Analytics for events that occurred during our Twitter data collection (18th August 2020 to 9th February 2022). Users accessed EIDP webpages directly (n=1,232), through social media (n=736), through referrals (linked from another website) (n=263), and organic searches (n=76). Twitter accounted for 98% of social media access to EIDP with the remainder from Facebook. The page view time series for the ten most viewed
events revealed a peak in page views associated with Sentinel-1 InSAR data tweets from @COMET_database for events with a deformation signal (n=6) (Figure 11). Three events did not have an associated data tweet and one had no deformation signal. The mean time between an earthquake and subsequent tweeting of InSAR data was 4.7±2.8 days, with the shortest gap being one day for the 2021 Mw 6.3 earthquake 'us7000df40' in Týrnavos, Greece (Table 4). The five most retweeted interferograms (Figure 10a) were amongst the most viewed EIDP event pages, of which the Mw 7.3 earthquake in Southern Qinghai, China
(21/05/2021, us7000e54r) had the most page views (n=1,231) (Figure 11, Table 4).



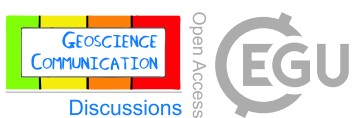

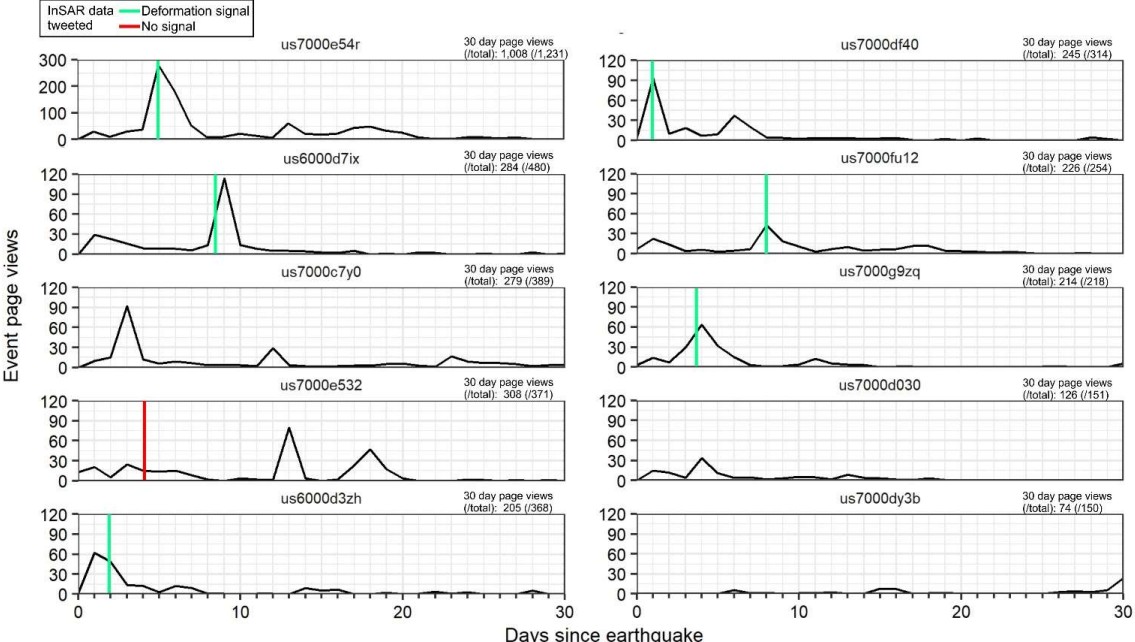

**Figure 11: Event page views for the ten most viewed earthquakes (Table 4). Vertical line shows the date that LiCSAR InSAR data were tweeted where applicable. (note y-axis change for the 2021 Southern Qinghai, China event (us7000e54r).**

**Table 4: Characteristics of the ten most viewed EIDP earthquake event pages, ordered by number of views (Figure 11). Earthquake details were obtained from the USGS (https://earthquake.usgs.gov/) on 4th April 2022**

| Location | Date and time (UTC) | USGS ID | Magnitude | Depth (km) |
|---|---|---|---|---|
| Southern Qinghai, China | 21/05/2021 18:04 | us7000e54r | 7.3 | 10 |
| 29 km SSW of Turt, Mongolia | 11/01/2021 21:32 | us6000d7ix | 6.7 | 10 |
| 13 km NNE of Néon Karlovásion, Greece | 30/10/2020 11:51 | us7000c7y0 | 7.0 | 21 |
| 25 km NW of Dali, China | 21/05/2021 13:48 | us7000e532 | 6.1 | 9 |
| 2 km WSW of Petrinja, Croatia | 29/12/2020 11:19 | us6000d3zh | 6.4 | 10 |
| 9 km W of Týrnavos, Greece | 03/03/2021 10:16 | us7000df40 | 6.3 | 8 |
| 63 km NNW of Bandar Abbas, Iran | 14/11/2021 12:08 | us7000fu12 | 6.4 | 10 |
| Southern Qinghai, China | 07/01/2022 17:45 | us7000g9zq | 6.6 | 13 |
| 32 km S of Mamuju, Indonesia | 14/01/2021 18:28 | us7000d030 | 6.2 | 18 |
| 8 km NNW of Dhekiajuli, India | 28/04/2021 02:21 | us7000dy3b | 6.0 | 34 |



### 3.2.3. Engagement with the COMET Volcano Deformation Portal


Development of the volcano InSAR portal tools has been informed by consultation with volcano observatory colleagues, including colleagues at Addis Ababa University, Ethiopia and Instituto Geofisico Escuala Politecnica Nacional, Ecuador. During consultations in 2021, feedback was positive from both groups about the ease of use of the portal interface and especially about the accessibility of displacement time series that were readily comparable to GNSS data or to independent InSAR processing results. In Ecuador, where InSAR data are already integrated into volcano monitoring practice, the portal was considered most useful as (1) an independent check on observatory results and (2) for interrogating timeseries, applying quick atmospheric corrections and comparing to topographic profiles. It was emphasised that it was critical for processing to be up to date for active volcanoes, in order for the portal to have a significant impact on monitoring. While Ecuadorian colleagues emphasised the importance of transparent explanations of the functionality of the different portal tools, Ethiopian colleagues considered general education about the potential of satellite radar in volcanology to be most important. This feedback is being integrated into plans for the development of the portal. Other developments of the volcano portal will focus on mitigation of 'fading signal' in long term time series, establishing near real-time processing that functions globally, and providing comprehensive guidance for interpreting LiCSAR data in multiple languages. The provision of Machine Learning tools will also be developed to incorporate a variety of complementary methods.



### 4 Discussion


Following the open access availability of SAR data acquired by the Sentinel-1 satellites (adopted for the Copernicus programme by the EU) and a transition towards open-source processing techniques, we aimed to evaluate the applications and engagement with InSAR products that are communicated online outside of traditional scientific publication outputs. First, we analysed InSAR geoscience communication through Twitter to determine the types of outputs communicated (e.g. text, images, videos), their data sources (e.g. Sentinel-1), geoscience applications (e.g. earthquakes, volcanoes), and level of user engagement. We then used our experience with the LiCSAR Sentinel-1 InSAR processing system to determine the extent to which this open access data provision aligned with Twitter InSAR communications.


### 4.1 InSAR Twitter

InSAR Twitter content was primarily aligned with natural hazards and disaster event response. However, peaks in Twitter activity were also associated with academic conferences (American Geophysical Union (AGU) and European Geosciences Union (EGU)), four earthquake events, news articles, and Twitter threads (one on subsidence and one on earthquake seasonality) (Figure 3). Analysis of the most commonly occurring words in the InSAR tweets revealed that InSAR was most associated with earthquake deformation (Figure 4a, d), likely reflecting the newsworthy status of these events drawing user engagement (e.g. Zawacki et al., 2022). In addition to earthquakes, applications of InSAR to volcanic deformation was also prevalent in our analysis, particularly, in the most retweeted InSAR tweets. Words such as 'dike', 'eruption', 'volcano', and







'magma' were present alongside specific events such as the volcano-tectonic events that occurred around the Icelandic Fagradalsfjall volcano on the 'Reykjanes' peninsula, and the 'Nyiragongo' eruption in Democratic Republic of the Congo (Figure 4c, e). Wrapped interferograms of these events were in the top ten most retweeted tweets (Figure 6).

The Sentinel-1 series comprised a pair of SAR satellites (Sentinel-1A launched in 2014 and Sentinel-1B in 2016, the latter which malfunctioned in December 2021), which provided a step change in provision and development of processing tools for open access radar data (Geudtner et al., 2014; Elliott, 2020). Our analysis of InSAR tweets reflect a prevalence of Sentinel-1 data usage as both the most commonly occurring hashtag (Figure 4b) and use as the data source in four of the top ten most retweeted InSAR tweets (Figure 6). For LiCSAR data, we observed greater retweeting of interferograms displaying an
earthquake deformation signal compared to those without (Figure 10), suggesting a degree of collaborative information filtering to amplify the presence of a positive detection. This aligns with other studies that found Twitter can facilitate rapid knowledge co-production that receives transparent, though unconventional, peer review through public Twitter discussions (Lacassin et al., 2020; Mendoza et al., 2010). Similarly, we found that *scientists* accounted for 38% of users within our defined Twitter InSAR network, though many of the *non-scientist* grouping likely represented Earth Observation experts in other
applications, including commercial industries. Whilst there is no guarantee that this network of users acts to positively filter InSAR data, for example, to mitigate common challenges such as misinterpretations of atmospheric effects as a deformation signal (for which we show a tweeted example of relating to Agung Volcano in Figure S2), other studies have shown that Twitter has been an useful platform for geoscience discussions and collating information on natural hazard events (Lacassin et al., 2020; Shugar et al., 2021; Ruan et al., 2022).

**4.2 LiCSAR data dissemination**

The availability of open access Sentinel-1 data has led to the development of semi- or fully-automated processing systems to produce InSAR information from the vast amount of data acquired each day. Scientific systems are largely focused on geohazards (e.g. Lazecký et al., 2020; Kennedy et al., 2021; Galve et al., 2017; Piter et al., 2021), whereas commercial systems cover a diverse range of applications including monitoring subsidence, mining activities, engineering projects, and geohazards.
Two commercial systems using Sentinel-1 and other data sources (@TRE_ALTAMIRA and @SatSenseLtd) were identified in the top tweeters InSAR network (Figure 4a). Open access LiCSAR data disseminated through an online portal and @COMET_database was also a top ten tweeter of InSAR data, specifically through automatic tweets of coseismic earthquake interferograms (Figure 10). LiCSAR InSAR data has been used in studies of earthquakes (Ganas et al., 2018), ground deformation (Morishita et al., 2020; Weiss et al., 2020; Chaabani et al., 2020), volcano deformation including machine learning
of deformation patterns (Moore et al., 2019; Anantrasirichai et al., 2018; Lloyd et al., 2018; Gaddes et al., 2018; Albino et al., 2019), and landslides (Burrows et al., 2019; Tsironi et al., 2022). The LiCSAR Portal had ~1,300 views per month (February 2021- February 2022) (Figure 7c) and user feedback suggested most respondents (77%, n=72) were academic users (Figure 8a). The primary link to data dissemination through Twitter was from the EIDP system that generates interactive web pages,



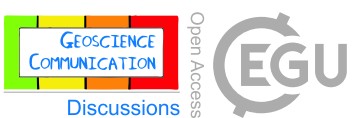

processes interferograms, and automatically tweets the outputs (Figure 9, 10, 11), thereby providing a resource for preliminary

interpretation and data download. Although LiCSAR data did not feature in the most retweeted InSAR content (Figure 6), the underlying data products in these tweets (wrapped and unwrapped interferograms) were the same as those generated by LiCSAR and these automatic tweets were associated with directing views to earthquake event pages (Figure 11). The most retweeted tweets generally featured supplementary data annotations and text explanations of the observed deformation signals, which are not a feature of the automated LiCSAR data dissemination. Similarly, 30% (n=24) of the LiCSAR survey

respondents identified *uncertainties* as a useful future product (Figure 8c), which would ideally be disseminated in parallel to other outputs to reduce misinterpretation of interferogram fringes for example, that are often related to atmospheric or topographic effects (Zhou et al., 2009).

### 4.2.3 Application of tailored online tools – the COMET Volcano Portal

Despite the prevalence of volcanic InSAR tweets (e.g. Figure 6), no responses were recorded from volcano observatories in

the LiCSAR user survey (Figure 8a). This may be in part because presentation of the COMET Volcano Portal thus far has been focused on targeted workshops (e.g., 'Online International Workshop on Volcano Monitoring Infrastructure on the Ground and In Space', Feb 18, 19, 22 2021, and the annual COMET InSAR Training Workshop) rather than social media. It may also reflect a preference of volcano-focussed users to engage with the volcano portal tools rather than downloading LiCSAR imagery.


Potential barriers to the uptake of InSAR data for volcano monitoring can be grouped into two broad classes, the first relating to the awareness of data availability and confidence in interpreting the data, the second relating to logistical challenges such as the human and computing resources required to process satellite data and challenges associated with internet bandwidth (Table 2). The COMET Volcano Portal minimises logistical issues by providing access to ready-processed InSAR data.

Although this is currently limited to selected volcanoes, the design of the portal is such that data for all volcanoes for which LiCSAR processing is reasonably complete could be included. As of November 2022, this totals 328 volcanoes that have either an ascending or descending frame ≥80% processed that fully covers a 25 km processing radius around each volcano, and 520 volcanoes that have both an ascending and descending frame meeting the same criteria. Data are clipped to the area of interest for individual volcanoes and resolution down sampling means that for example the timeseries data are served over a standard

web browser in small files sizes (typically several megabytes). Dealing with the second barrier of data awareness and confidence requires an approach incorporating a communication strategy. This could include Tweeting examples of portal content and promoting the portal at conferences and through local observatory networks. We consider regular communication of volcano portal updates to be a better strategy than automatic direct social media responses to eruptions or unrest, due to the potential for eclipsing or distracting from the official communications from local volcano observatories and civil protection.

We also consider further direct engagement with volcano observatory scientists to be critical, both for making the portal as applicable and transparent in its operation as possible, and for making sure that communication of how to use the online tools



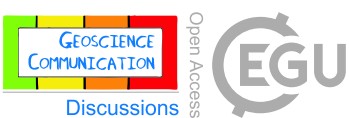

reaches the most appropriate audiences. Demonstrating the applicability and reliability of automatically produced InSAR data for volcano monitoring would benefit from comparisons with bespoke processing and ground truthing, both of which also require collaborations with local volcano observatories and researchers. These are also aided by engagement with international

initiatives such as  the CEOS Volcano Pilot/Demonstrator (Pritchard et al., 2018) or the Global Volcano Monitoring Infrastructure Database (GVMID). Continual automated processing that is integrated into existing monitoring networks therefore offers opportunities to inform broader goals of preparedness and disaster risk reduction.

**4.3 InSAR communications and disaster risk reduction**

Overall, InSAR Twitter was orientated towards natural hazards and specifically disaster or event response, rather than
preparedness or disaster risk reduction, which reflects the current affairs nature of the Twitter platform (Petrovic et al., 2013; Murthy, 2011; Acar and Muraki, 2011). Here, InSAR contributes top-down situational awareness to these natural hazard events, usually through providing satellite observations of ground deformation (e.g. Figure 6, 10a), in some cases on interactive maps that could be used alongside 'on the ground' mapping (Figure 9b). By comparison, bottom-up aggregation and analysis of 'on the ground' crisis-communication tweets are shown to be effective at identifying the immediate aftermath of events
such as earthquakes, and providing the first accounts of impacts (Vieweg et al., 2010; Acar and Muraki, 2011; Earle et al., 2011). We found that peaks in InSAR tweets corresponded earthquakes (Figure 3) and ongoing volcanic events (Figure 4c), where InSAR products were used to reveal the spatial patterns of deformation. For the automated @COMET_database tweets, data were acquired, processed, and tweeted within 4.7±2.8 days of the earthquake event, which may limit applicability of the data in the immediate response effort. It would therefore be useful to investigate the links between data dissemination,
knowledge building over Twitter, and local event response to determine the specific applications and benefits to disaster management (Figure 12). This is particularly relevant when considering the coupling between open access data, derived products, related uncertainties, and quality interpretation, which are all necessary for effective communication.



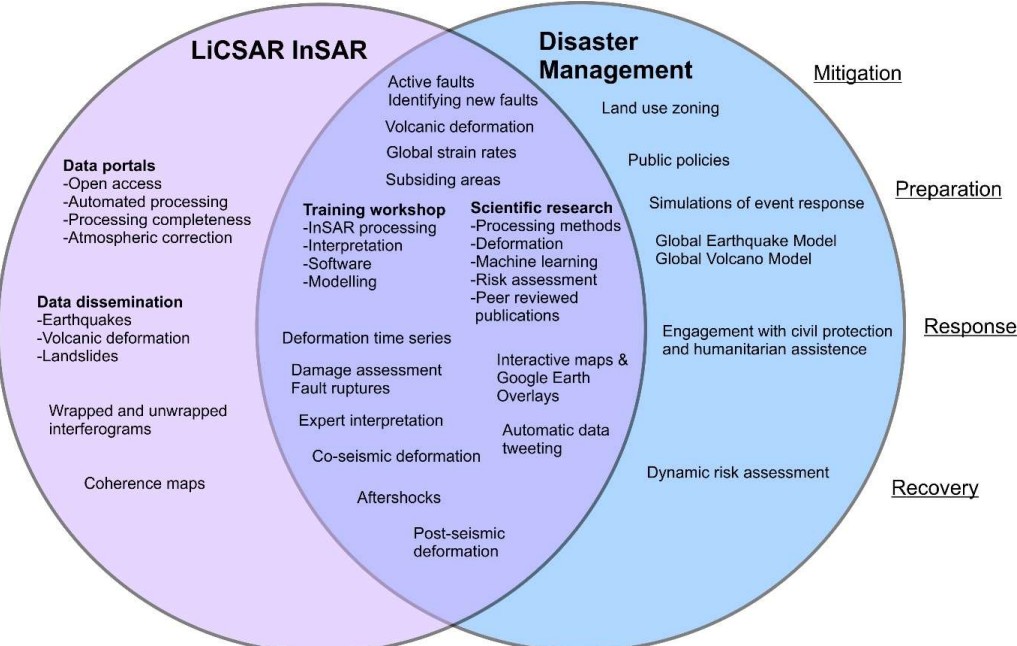

**Figure 12: InSAR intersection with natural hazard disaster management.**


In addition to event response, we observed tweets related to longer timescale events such as ground subsidence. For example, a Twitter thread on subsidence in Mexico City due to urbanisation and groundwater extraction did not directly include InSAR data (Figure 6c). Explanatory Twitter threads are effective means of knowledge building and dissemination (Lacassin et al., 2020) and could therefore be used to incorporate InSAR analysis into the broader and longer-term disaster management cycle

including reduction, readiness, response, and recovery (Elliott, 2020; Joyce et al., 2009) (Figure 12). Twitter threads unrelated to disaster events, for example on long-term measurements of active faults, offer one mechanism to balance the current focus on post-disaster response applications and media reporting, which increasingly embed Twitter threads within their news articles and therefore increase public exposure (Broersma and Graham, 2013).

Pre-processed and open access InSAR data such as produced by LiCSAR could support disaster risk reduction strategies, for example through long-term strain rate mapping over active faults to improve seismic hazard assessments (Lazecký et al., 2020; Stevens and Avouac, 2021), or monitoring regions prone to landslides (Novellino et al., 2021; Rott and Nagler, 2006) or volcanoes (Ebmeier et al., 2018). Similarly, InSAR can reveal urban ground deformation (Morishita, 2021) due to groundwater extraction or the weight of a city itself (Parsons, 2021). Therefore, several opportunities exist to integrate InSAR into disaster

management strategies and broader UN Sustainable Development Goals to develop sustainable cities with reduced risk from





natural hazards (UNISDR, 2015; UN, 2020). Increased uptake and dissemination of InSAR data has followed the breakdown of barriers to data accessibility, including open access data and processing chains; however, improved metadata detailing processing methods and uncertainties are still required to maximise applications of InSAR data and minimise misinterpretations due to radar data complexities (Table 1). For example, evaluating apparent deformation signals alongside

weather models is often essential at volcanoes to avoid incorrectly inferring a deformation signal (e.g. Figure S2). Integration of InSAR datasets into cloud-based analysis platforms and portal such as Google Earth Engine or the European Space Agency's Geohazard Exploitation Platform could further facilitate analysis with other data sources, and offers cloud computing capabilities for users lacking local resources (Gorelick et al., 2017), which is potentially advantageous for embedding InSAR data into disaster management agencies and volcano observatories (Table 2).

**4.4 Future design and dissemination of LiCSAR products**

Our analysis of InSAR communications has shown that Twitter is a valuable tool for disseminating InSAR data to scientific and other Twitter users (Figure 5) and that current communications are predominantly focussed on natural hazard response (Figure 3, 4). LiCSAR products are targeted at geoscience researchers monitoring actively deforming regions of the Earth's surface, which includes production of interferograms in response to earthquakes and could similarly be applied during volcanic

eruptions (Lazecký et al., 2020). Whilst the products are relevant for improving scientific understanding of events and processes, operational crisis response, and disaster mitigation, they are not produced with a real-time disaster response mandate. Therefore, improving the dissemination of LiCSAR products for use in crisis response requires links with local or national advisory agencies, for example the British Geological Survey in the UK. The type of disaster (e.g. earthquake or volcanic eruption) and event-specific sensitivities require consideration to avoid hindering response efforts and to ensure

effective and ethical communication (Newhall, 1999). Since ongoing hazard events have specific contexts and require local considerations beyond the scope of COMET LiCSAR, data dissemination is best channelled to organisations that can effectively use the InSAR data and link with civil protection agencies. This could involve additional signposting of data portals (e.g. EIDP Figure 9) and provision of guidance text and flowchart diagrams for data interpretation. Longer term integration of InSAR data into disaster mitigation (Figure 12) could also benefit from expert led Twitter threads discussing and demonstrating

the capabilities and limitations of InSAR for specific applications.

Communication of uncertainties in LiCSAR InSAR data are currently lacking and were noted as a desired future product in a user survey (Figure 8c). Data processed with higher levels of corrections (e.g. atmospheric) particularly using a single methodology, do not always improve the end result and could cause misinterpretations (Table 1). Additionally, other spatially

variable (e.g. depending on land cover) biases can accumulate in time series analysis of interferograms (Maghsoudi et al., 2022; De Zan et al., 2015). Therefore, effective communication of processing steps is required to avoid instilling a false sense of confidence in derived outputs. Higher level processing steps require waiting for the necessary correction data and further processing time, which may delay use of the data. Therefore, there is a trade-off between data timeliness and appropriate level

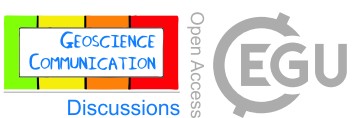

of processing, which is variable between hazard types and individual events. For example, there is merit in producing

interferograms of earthquake events with a large deformation signal before atmospheric corrections (Figure 10a), whereas the signal for smaller or deeper events could be obscured.

We identify the following recommendations as a pathway for LiCSAR development and InSAR communications:

**Recommendation (1):** Metadata documentation and version control of the processing steps used to derive data outputs should be provided for each LiCSAR frame. Where feasible, data should be provided at each level of processing (e.g. pre- and post-atmospheric correction) when corrections are applied. Corresponding difference maps would also reveal the spatially variable nature of the applied corrections. Metadata is essential for consistent and reproducible analysis and to allow third-party users or organisations to derive higher level products from InSAR datasets such as LiCSAR.


**Recommendation (2):** Development of new and existing products should be accompanied by clear communication of the uncertainties associated with each stage of processing and guidance for interpretation. This is important for higher-level products such as time series displacement data, where unresolved uncertainties and biases may be present.

**Recommendation (3):** Develop a protocol for generating priority LiCSAR data in response to volcanic activity, and making sure that COMET Volcano Portal outputs are accessible to volcano observatories.

**Recommendation (4):** Data intercomparisons with non-LICSAR processing streams and methodologies such as Alaska Satellite Facility's HyP3 (Meyer et al., 2017) and the GeoHazards Exploitation Platform (Galve et al., 2017) should be
encouraged to help refine processing methodologies and quantify uncertainties.

### 5 Conclusion

In this study, we evaluated the dissemination of ground deformation measurements from InSAR through Twitter in relation to sources of open access data. Twitter activity was primarily associated with natural hazard response, specifically following earthquakes and volcanic activity, where InSAR measurements of ground deformation were tweeted. For earthquake events in
the LiCSAR EIDP system, data were acquired, processed, and tweeted within 4.7±2.8 days (shortest was one day). Applications of open access Sentinel-1 data prevailed in the InSAR tweets and were applied to volcanic and earthquake events in the most engaged with (retweeted) content. Twitter InSAR dissemination was primarily outreach to a non-scientific audience (62%), including industry Earth Observation experts. LiCSAR data were widely accessed through the online portal and disseminated following major earthquake events to reveal ground deformation. However, further work is required to integrate
dissemination of InSAR data into longer-term disaster risk reduction strategies and broader UN Sustainable Development



Goals to develop sustainable cities, in addition to the prevailing emergency response applications. This requires tailoring the online analysis tools that are founded on open access data to meet a diverse range of end user needs, including long-term monitoring of tectonic and volcanic activity that can better inform risk reduction strategies in the context of an urbanising global population. Effectively communicating uncertainties is also critical given the often complex interpretability of InSAR

products.

**Data availability**

The data used to support the findings and results of this study are available from the LiCSAR Portal (https://comet.nerc.ac.uk/comet-lics-portal/) and in the supplementary information. Twitter data are publicly accessible through Twitter (https://twitter.com/).


**Author contributions**

All authors have read and agreed to the published version of the manuscript. CSW designed the concept with JRE and SKE. CSW performed the analysis, prepared the figures, and wrote the manuscript with input from all authors. JB provided the summary information of a survey on barriers to uptake of satellite data identified by volcano observatories.


**Competing interests.**

The authors declare that they have no conflict of interest.

**Financial support.**

This work was supported by the Centre for the Observation and Modelling of Earthquakes, Volcanoes and Tectonics (COMET) in the United Kingdom, and the Natural Environment Research Council through the Looking into the Continents from Space (LiCS) large grant (NE/K010867/1). CSW is funded by COMET and the GCRF Urban Disaster Risk Hub 'Tomorrow's Cities' (NE/S009000/1). JRE acknowledges support from the Royal Society through a University Research Fellowship (UF150282). SKE is supported by a NERC Independent Research Fellowship (NE/R015546/1).


**Acknowledgements**

COMET is the NERC Centre for the Observation and Modelling of Earthquakes, Volcanoes and Tectonics, a partnership between UK Universities and the British Geological Survey. LiCSAR uses JASMIN, the UK's collaborative data analysis environment (http://jasmin.ac.uk). This study contains modified Copernicus Sentinel data (2015-2022). We thank Elias Lewi

Teklemariam, Patricia Mothes, Pedro Espín Bedón, Marco Yépez, Marco Córdova and Santiago Aguaiza for feedback during the development of the COMET Volcano Deformation Portal.



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
