# Peer review of "Strategies for improving the communication of satellite-derived InSAR ground displacements"

_Geoscience Communication, 2022_

## Referee Comment (RC1)

Referee Comments:

*Strategies for improving the communication of satellite-derived InSAR ground displacements*

**General Comments:**
This paper investigates the distribution and sharing of open access InSAR data products amongst scientists and non-scientists on Twitter and other platforms. Researchers also explore potential issues with the sharing of this type of data, responses to existing InSAR data outlets and their use for different types of hazards. They briefly discuss a web portal they developed and feedback from the international community on what is needed. This paper has the potential to be a valuable contribution to the literature to better understand how different groups use InSAR data and to provide useful recommendations for the future.

However, before I can recommend publication, I encourage the authors to distil their key messages and remove extraneous information, perhaps even splitting the information provided here into two papers. The first paper could examine aims 1 and 2, "to evaluate (1) who interacts with disseminated InSAR data, (2) how the data are used" and the second paper could explore aim 3, "to discuss strategies for meaningful communication and dissemination of open InSAR data" where authors could explore issues with interpretation of InSAR products and outline recommendations. Combining all three aims into this one paper makes it hard for the reader to pull out important insights as there is so much information presented here.

Specifically, there are too many multi-part figures included in this manuscript. As a reader, it was hard to determine which were most relevant for the findings and where to focus. I would encourage the authors to do a thorough review of the figures to determine which are most necessary to make their main points. Below I have included some suggestions for doing this.

I would also encourage the authors to include "road map" sections at the beginning of the major sections where you summarize what is coming in the following section to help orient readers. You employed several different avenues for investigating this topic. By including more roadmap sections readers can be more aware of what is coming and how the paper is organized. I found myself getting lost at times.

This is important work and I commend the authors for using novel methods to better understand how SAR data and products are being used and by whom. I hope the comments below are helpful in considering how to best communicate the valuable research you are doing.

**Specific Comments:**

**Title:** I recommend the authors reconsider their title and narrow the scope. The abstract mainly focuses on the investigation of Twitter posts and online InSAR data portals and the title should reflect that. Given my previous recommendation to split the paper in two, a revised title could focus on the data mining from Twitter and other platforms while the second could focus more on barriers and recommendations. See below for recommendations.

- Current title: Strategies for improving the communication of satellite-derived InSAR ground displacements
- Example title for paper 1: Examining the Use and Users of satellite-derived InSAR data for natural hazard events through Twitter and online data portals
- Example title for paper 2: Assessing barriers to the communication of satellite-derived InSAR data use and recommendations for improvement

**Abstract:**
As it is written, the abstract summarizes some of the content of the paper related to data mining from twitter and other platforms. However, it leaves out key results of aim 1 – who is interacting with InSAR data? What types of information are they sharing? I recommend adding this information to the abstract.

**Introduction:**
1.1.
A general comment for this section is to simplify it. There is a lot of fine detail and discussion about the history and specifics of SAR and other satellite data. I think you can summarize it more concisely to include what SAR and InSAR data are, how they are different than previous sat imagery, and their uses.

Line 36. *"[The] availability of open access Sentinel-1 data over the last ~8 years, in addition to other emerging SAR satellites, creates an opportunity for making SAR and InSAR data accessible and useable to non-experts."*
I recommend moving this sentence to be the opening sentence of this section as it better summarizes this section than the current intro sentence.

Line 40-42. *"Optical…(Fowler, 2013)"* I am not sure you need this level of detail. I recommend jumping right into specifics about interpreting SAR data as it compares to other satellite imagery.

Table 1 and Figure 2.
- How were the interpretation challenges determined? Were these from personal experience? Assumed? Please clarify how you came to these conclusions.
- It would be helpful if the interpretation challenges were simplified/connected with implications for users. Right now, it is not clear what the implications of these challenges are. For example, under Coherence, one challenge is, "The magnitude of coherence change is a function of time." Can you explain why this is an issue? What would this cause a non-expert user viewing this information to do? Could you revise this column to include this information?
- Additionally, I believe Table 1 would be more powerful if it were combined with the images in Figure 2 so readers can look at the images while they read about the potential issues. I recommend using the same landscape area with the four different products, if

possible, to allow viewers to easily compare across product types. (see example of Combined Table 1 and Figure 2. on next page)

Line 121. Newhall, 1999 reference should be in addition to newer references on this topic. See West et al., 2021 and references section for additional literature.

Line 137. You provide examples of "outreach," but because this paper is using the term in a specific way and for analysis, it would be helpful to provide a specific definition of 'outreach.' For example, "In the paper we use the term "outreach" to represent interactions where individuals we classify as scientists do x, y z… (be explicit with what you classified as outreach and why).

Combined Table 1 and Figure 2.

| Example |  |  |  |  |
|---|---|---|---|---|
| Product | Coherence | Wrapped interferogram | Unwrapped interferogram | Time series |
| Description | A measure of stability of the scattering surface through time. | Ambiguous measure of apparent surface deformation in cycles of phase (fringes). | Cumulative measure of apparent surface deformation in the satellite line of sight. | Temporal deformation change derived from a network of interferograms. |
| Key Potential/ Observed Interpretation Challenges/ Implications | • … | | | |

**Data and Methodology:**
I recommend adding a "roadmap" paragraph after section 2 and before 2.1 where you describe the various methods you used and what they were for. Then in the subsections you can expand on the finer details.

2.1.

Line 145. I recommend adding a topic sentence to summarize the different analyses and data types pulled from Twitter before jumping into specifics of analysis packages.

Line 146. "Twitter data are increasingly used in scientific analysis." I don't think this is needed since data mining in Twitter are common practice these days.

Line 165. In your description of how you classify scientists and non-scientists, can you clarify where government agencies that focus on science would be classified? I am guessing as "science," but government agencies are listed under the "non-scientist" category.

2.2
2.2.1 – I like that you included ethical considerations for Twitter users, but perhaps this can go toward the end of the paper in a "limitations and other considerations" section?

Lines 1 189-190. *"Responses were also solicited by COMET members through peer networks and existing partnerships. COMET membership was asked as a survey question to distinguish these responses."*
Did you send COMET members the feedback survey directly? Were they asked only one survey question? Was this upon signing up for membership or for current members? Please be more explicit here with how you collected this information.

2.4
The content in this section seems beyond the scope of this paper as it is not summarizing users of InSAR but rather gathering and providing recommendations as well as discussing a new tool that address those recommendations. With only 10 survey responses, I am not sure it is worth reporting the results here. These responses are valuable, but would carry more weight if there were more and solicited from both developing and more established countries to see similarities and differences in their data needs. Since this paper examines general use of InSAR, I am not sure if this narrowing of users to developing countries/overseas territories is appropriate in this paper. Additionally, authors go into a specific tool they developed with these recommendations which also seems to be outside the scope of this paper. I recommend removing this section.

Line 216. This is getting into results from the 10 questionnaires and should not be in the methods section.

**Results:**

3.1
Line 264. Missing comma between 2021 and respectively.

Figure 3. Recommend adjusting caption into to read "Timeline of InSAR tweets from September 2020 to February 2022…."

Figure 4. Though interesting, I think word clouds a.- c. are unnecessary and can be removed. Bar graphs d and e are more helpful for understanding.

3.2
Figure 7. This figure is very busy and hard to see what the most important aspect is. It seems like 7c is most relevant for this paper. I am not sure a. and b. are helping readers interpret results. If they are interested in what the portal looks like, they can go to the website.

Line 362. I am guessing the data frame is the region a user is pulling the data from. Can you please briefly describe what this is here for those who may be less familiar?

Figure 8. Pie charts are best for responses with 3 or less categories. I recommend adapting this into bar charts similar to a-c for easier interpretation.

Figures 9 and 10. Are these full images needed? This paper has many figures and I wonder if they are all necessary for in-text reference? Could these be added as supplements or paired down to only include the most important pieces? As a reader, it is somewhat overwhelming to tease out important aspects of these multipart figures.

Figure 11 and Table 4: Similar to a previous comment. Could the text from Table 4 be added to the top of the panels in Figure 11 so readers could easily identify the event that ties to the page view count without having to flip between the table and figure? See example below:

[Figure]

3.2.3.
This section goes into detail of a specific portal and changes made from recommendations from a small sample of colleagues. Again, I think because this goes into such specifics it is beyond the scope of the paper outlined in the abstract and should be included in a separate manuscript.

**Discussion:**
4.1
Line 434 – 437. Nice topic sentence for this section.

In your research did you find that non-expert users were misinterpreting the data or information given the potential interpretation challenges for the SAR/inSAR data products? It would be nice to see some discussion of this.

4.2
Line 491 – 512. This discussion may be outside of the scope of this paper or could be reduced in size as it gets a bit into the weeds.

4.3
Figure 12. I would recommend updating this figure. To make clear how the stages of disaster management relate to InSAR data product. Right now, they are sitting on the outside of the figure and it is not clear what their placement means or how it relates to the SAR information.

4.4
Should this section name also include InSAR, "4.4 Future design and dissemination of LiCSAR **and InSAR** products" since the recommendations apply to both?

Line 585-600. I would recommend bolding each recommendation so it stands apart from the description of the recommendation.

**Conclusion:**
What is written in the conclusion summarizes part of what is discussed in the paper. Apart from analyzing Twitter, authors also looked at surveys, and other analytics from data platforms, and discussed a portal that was developed, as well as international feedback they received. If all of this is to remain in the paper, it should also be mentioned in the conclusion. However, my recommendation remains to refine the paper to fit what is currently in the conclusion and take out the additional pieces to develop a second paper.

I recommend including a limitations section for this paper. The Twitter ethics statement can also be included with this section.

---

## Referee Comment (RC2)

General Comments

I think this is a very valuable subject for study, and I think that the authors have done the subject justice in this comprehensive and thoughtful manuscript. The work is coherent and well-reasoned throughout, with consideration of the multiple stakeholders that use InSAR data and specific areas where the authors show they have considered how this data may be processed and communicated ethically. The abstract is clear and covers the aim, results, and most of the important implications of the study. The manuscript is well-structured and easy to read throughout; the authors write fluently. Figures are easily understandable and clearly illustrate the relevant points in the manuscript, while the tables give a good summary of the meaning behind and application of different InSAR products. Well done to the designers, and to all involved in this study!

I have a couple of suggestions to make that I hope will further improve the quality of the manuscript. First, the abstract highlights the current challenge in communicating uncertainties with these data. I think this is an important and interesting challenge, but in the manuscript this section was shorter than I'd expected. It appears in the Conclusions, which references Figure 8c that specifies users strongly desire uncertainties included in future products. Do you have more information as to what uncertainties these might include? Adding 1-2 sentences of greater detail on uncertainties would bolster these conclusions – and given you end the abstract with them, they do feature quite prominently to a reader.

The second suggestion I would make would be to include a short section on the limitations of this study. I was surprised to not find any mention of the limitations of using Twitter in the first place. As the authors note, Twitter is used extensively by academic and industry scientists, and scientists in this study express that they want to communicate these data with non-scientists (i.e., outreach). But how much is Twitter used by these non-scientists? Exploring what platforms are used by these stakeholders should **obviously** not be added to this manuscript – it's another study entirely. Given that at least some of your colleagues (e.g., in Ethiopia) consider general education about the potential of satellite radar in volcanology to be important, I think there's scope to mention how Twitter is used only by some stakeholders (for instance, civil protection in the place I work in often use Instagram and Facebook for communications). Including a section on study limitations could also include a brief consideration on misinformation related to communication of InSAR data. I am uncertain as to how big a problem it is, but certainly misinformation around other volcanic hazards shared on social media. If your results do not give any evidence to contribute to the discussion on misinformation around InSAR data, perhaps it would be worthwhile to signpost the possibility of misinformation so that it might be explored in a subsequent study.

My third suggestion is regarding Sections 2.4 (Data: COMET Volcano Deformation Portal) and 3.2.3 (Results: Engagement with the COMET Volcano Deformation Portal). These sections are fascinating and I think deserve more attention. However, I see a certain challenge in that the participants in each of these sections are relatively few (10 respondents in Section 2.4, and two responses in Section 3.2.3). Furthermore, they constitute a different kind of engagement to Twitter, in that responses and feedback are much more individual. I think the barriers to uptake of satellite data you present in Table 2 is very valuable, and I would like to see the engagement with this portal and future directions explored in more depth than what is

presented in Section 3.2.3. With all that said, I recognize that this paper is already long at 31 pages, and that adding more detail to Sections 2.4 and 3.2.3 would make it even longer. Do you have enough data to present these sections as part of a separate, shorter second paper? That would be my recommendation. I think it would have the dual benefit of honing the focus of this excellent paper and giving sufficient space and respect to the interesting but too-short results presented in these two sections.

Specific Comments

**COMET Volcano Deformation Portal and LiCSAR portal** – AMAZING that these tools are out there, and that they are being used to engage with countries with active volcanoes worldwide!

**Lines 125 – 128:** these are very important and interesting questions. I read these as questions you might be answering in your study, and so was a little put out that they didn't appear again. Could you perhaps add a sentence to clarify that these are not the research questions you will be answering in this study but will instead be answered in another paper?

**Line 137:** I think it's very interesting that you have distinguished in-reach from outreach, to see how much data moves outside the bubble. Great! Suggestion to add "… and the types of data that are communicated *in each case"* to emphasize that you will be studying the different data types for in-reach vs outreach.

**Line 146**: on excluding retweets: I imagine this reduces the dataset to a manageable size, but do you then remove the possibility of exploring misinformation through images attributed to other events? I don't think you should add in data RTs, but I suggest that you could signpost the possibility of exploring retweets to identify a direction for future studies – that is, if misinformation is an issue for InSAR data communicating natural hazards (it certainly is for other types of data relating to volcanic hazards).

**Line 155**: "(2) the tweeters of the top ten most retweeted tweets." Congratulations on this excellent tongue-twister.

**Lines 155 – 156**: I find the wording of this pretty confusing. Can I clarify: *"top ten InSAR tweeters, (1) ranked by number of tweets, and (2) the tweeters of the top ten most retweeted tweets"* – I read this to mean that (1) are the ten accounts with the largest number of tweets in your study period, and (2) the accounts from which the ten tweets with most RTs come from during our study period. Is this correct? If not, I would suggest rewording to make this clearer.

**Line 184:** starting when in 2020 for Wordpress?

**Line 185:** can you suggest a reason why the two are not directly compatible?

**Line 210:** what language did you send the questionnaires in? Please specify in text.

**Lines 214 – 215:** the number of volcanoes … amazing! Was this Guatemala? I hear a different answer every time I ask.

**Table 2:** I think this is an excellent table. Thoughts:
- Is there any significance to the different colours of the table columns? White looks like scientific problems, grey like capacity or resources. Please explain the difference in your table caption.
- This is perhaps my ignorance, but "automated processing" to me reads as building capacity for automated processing within institutions themselves. Wouldn't this require more computing capacity and high-speed internet, which you state is outwith the capacity of these institutions? Would "ready-processed and analysed data" be a more precise way of describing what you mean? (I lifted this from your text, which I think describes the challenges neatly.

**Lines 414 – 418:** very, very interesting about the large difference between Ecuador and Ethiopia about the use of the portal – former in-reach (QC) and latter, outreach. I would love to read more about this, especially if and how you plan to continue seeking feedback from portal users and then implementing it. However, as I mention in my General Comments, this seems to be a different direction from the majority of the paper. Perhaps consider removing this section and sending to another one.

**Lines 502 – 504:** "*We consider regular communication of volcano portal updates to be a better strategy than automatic direct social media responses to eruptions or unrest, due to the potential for eclipsing or distracting from the official communications from local volcano observatories and civil protection*." – this is a very thoughtful and responsible stance for doing ethical science. Congrats!

**Lines 514 – 527:** "*Overall, InSAR Twitter was orientated towards natural hazards and specifically disaster or event response, rather than preparedness or disaster risk reduction, which reflects the current affairs nature of the Twitter platform (Petrovic et al., 2013; 515 Murthy, 2011; Acar and Muraki, 2011).*" – this is very interesting, and as I wrote in the General Comments, I think it's important to consider that people involved in preparedness or DRR do not use Twitter as their primary platform. I think your following sentence: "*It would therefore be useful to investigate the links between data dissemination, knowledge building over Twitter, and local event response to determine the specific applications and benefits to disaster 525 management (Figure 12)*" is very valuable. Have you ever seen a study that does this? If so, I would recommend referencing it here. If not, I think you could include this_ as a Recommendation. Perhaps after your recommendations for LiCSAR development and InSAR communications, you could include this as a separate Recommendation for integrated communication of natural hazards? It would be a highly valuable but complicated and intricate study.

Technical Corrections

- **Line 59:** "in addition *or* anthropogenic signals" – should be "to", perhaps?
- **Line 74:** "towards or *award* from the satellite" – should be "away", perhaps?

- **Line 126:** "consideration for teams that remote to the event" – add "are" before "remote"
- **Line 174:** "*though* acceptance of Twitter's user terms" – should be "through"
- **Line 175:** "Tweets" – lower-case
- **Lines 246 – 248:** you're missing a bracket to close the "(this …"
- **Line 359:** remove comma after "interferograms"
- **Line 440:** remove comma after "particularly"
- **Line 536:** "offer *one* mechanism" – this is ambiguous – what is that mechanism? Or do you mean the threads are the mechanism? In which case, I'd switch the "one" to "a".

---

## Author Comment (AC1)

**Response to Geoscience Communication Reviewer 1**

**Manuscript:** https://doi.org/10.5194/gc-2022-15
**Review**: https://doi.org/10.5194/gc-2022-15-RC1
**Response date:** 27-March-2023

Dear Reviewer,
Thank you for your positive feedback and comments on our manuscript. We look forward to incorporating your suggestions in a revised manuscript submission. Please find our response to your specific comments in **bold**, with revised text in green.

Sincerely,
C. Scott Watson (on behalf of all co-authors)

**General comments:**

This paper investigates the distribution and sharing of open access InSAR data products amongst scientists and non-scientists on Twitter and other platforms. Researchers also explore potential issues with the sharing of this type of data, responses to existing InSAR data outlets and their use for different types of hazards. They briefly discuss a web portal they developed and feedback from the international community on what is needed. This paper has the potential to be a valuable contribution to the literature to better understand how different groups use InSAR data and to provide useful recommendations for the future.

However, before I can recommend publication, I encourage the authors to distil their key messages and remove extraneous information, perhaps even splitting the information provided here into two papers. The first paper could examine aims 1 and 2, "to evaluate (1) who interacts with disseminated InSAR data, (2) how the data are used" and the second paper could explore aim 3, "to discuss strategies for meaningful communication and dissemination of open InSAR data" where authors could explore issues with interpretation of InSAR products and outline recommendations. Combining all three aims into this one paper makes it hard for the reader to pull out important insights as there is so much information presented here.

Specifically, there are too many multi-part figures included in this manuscript. As a reader, it was hard to determine which were most relevant for the findings and where to focus. I would encourage the authors to do a thorough review of the figures to determine which are most necessary to make their main points. Below I have included some suggestions for doing this.

I would also encourage the authors to include "road map" sections at the beginning of the major sections where you summarize what is coming in the following section to help orient readers. You employed several different avenues for investigating this topic. By including more roadmap sections readers can be more aware of what is coming and how the paper is

organized. I found myself getting lost at times.

This is important work and I commend the authors for using novel methods to better understand how SAR data and products are being used and by whom. I hope the comments below are helpful in considering how to best communicate the valuable research you are doing.

**We have decided not to split this paper into two since we do not believe a second paper would include enough content. We also believe that including the Volcano Portal adds valuable insight into how such portals should be developed in response to user input. However, we recognise that both reviewers identified some of the Volcano Portal information as a divergence from the main theme of the paper and have therefore reworked and streamlined all volcano portal sections to integrate this work, including the important but small subset of feedback. We have reduced multipart figures in several places and have included introductory roadmaps in response to your comments. We have also reduced the overall length of the paper where possible.**

**Specific comments**

Title: I recommend the authors reconsider their title and narrow the scope. The abstract mainly focuses on the investigation of Twitter posts and online InSAR data portals and the title should reflect that. Given my previous recommendation to split the paper in two, a revised title could focus on the data mining from Twitter and other platforms while the second could focus more on barriers and recommendations. See below for recommendations.
• Current title: Strategies for improving the communication of satellite-derived InSAR ground displacements
• Example title for paper 1: Examining the Use and Users of satellite-derived InSAR data for natural hazard events through Twitter and online data portals
• Example title for paper 2: Assessing barriers to the communication of satellite-derived InSAR data use and recommendations for improvement

**Although we have not split this paper into two, we recognise the value in adding those extra details to the title so have changed to:**
**Strategies for improving the communication of satellite-derived InSAR data for geo-hazards through the analysis of Twitter and online data portals**

Abstract:
As it is written, the abstract summarizes some of the content of the paper related to data mining from twitter and other platforms. However, it leaves out key results of aim 1 – who is interacting with InSAR data? What types of information are they sharing? I recommend adding this information to the abstract.

**We have now added the percentage breakdown of scientist vs non-scientists to the abstract which is then followed by examples of the types of InSAR data that were commonly tweeted in response to earthquakes and volcanic activity.**

We found that the InSAR Twitter community was primarily composed of non-scientists (62%), although this grouping included Earth Observation experts in applications such as commercial industries. Twitter activity was primarily associated with natural hazard response, specifically following earthquakes and volcanic activity, where users disseminated InSAR measurements of ground deformation, often using wrapped and unwrapped interferograms.

Introduction:
1.1.
A general comment for this section is to simplify it. There is a lot of fine detail and discussion about the history and specifics of SAR and other satellite data. I think you can summarize it more concisely to include what SAR and InSAR data are, how they are different than previous sat imagery, and their uses.
**We have simplified this section as suggested to shorten the introduction and clarify the key points.**

Line 36. *"[The] availability of open access Sentinel-1 data over the last ~8 years, in addition to other emerging SAR satellites, creates an opportunity for making SAR and InSAR data accessible and useable to non-experts."*
I recommend moving this sentence to be the opening sentence of this section as it better summarizes this section than the current intro sentence.
**We have moved this sentence as suggested.**

Line 40-42. *"Optical…(Fowler, 2013)"* I am not sure you need this level of detail. I recommend jumping right into specifics about interpreting SAR data as it compares to other satellite imagery.
**We have removed these two sentences as suggested.**

Table 1 and Figure 2.
• How were the interpretation challenges determined? Were these from personal experience? Assumed? Please clarify how you came to these conclusions.
**There were derived from the experiences of the authorship team. We have now changed this column header to read 'Interpretation challenges based on the authors' experiences.**

• It would be helpful if the interpretation challenges were simplified/connected with implications for users. Right now, it is not clear what the implications of these challenges are. For example, under Coherence, one challenge is, "The magnitude of coherence change is a function of time." Can you explain why this is an issue? What would this cause a non-expert user viewing this information to do? Could you revise this column to include this information?
**We have now added additional explanation to the points that did not specify the implications of the challenge.**

• Additionally, I believe Table 1 would be more powerful if it were combined with the

images in Figure 2 so readers can look at the images while they read about the potential issues. I recommend using the same landscape area with the four different products, if possible, to allow viewers to easily compare across product types. (see example of Combined Table 1 and Figure 2. on next page)

**We agree that this combination could help the reader; however, the detail of the images would likely be compromised and therefore still require a separate figure, and it is not clear if such a table-figure combination is permissible within the journal so we decided not to revise this figure.**

Combined Table 1 and Figure 2.

| Example | | | | |
|---|---|---|---|---|
| Product | Coherence | Wrapped interferogram | Unwrapped interferogram | Time series |
| Description | A measure of stability of the scattering surface through time. | Ambiguous measure of apparent surface deformation in cycles of phase (fringes). | Cumulative measure of apparent surface deformation in the satellite line of sight. | Temporal deformation change derived from a network of interferograms. |
| Key Potential/ Observed Interpretation Challenges/ Implications | • … | | | |

Line 121. Newhall, 1999 reference should be in addition to newer references on this topic. See West et al., 2021 and references section for additional literature.

**We have added additional references from Pallister et al., (2019) and West et al., 2021.**

Pallister, J., Papale, P., Eichelberger, J., Newhall, C., Mandeville, C., Nakada, S., Marzocchi, W., Loughlin, S., Jolly, G., Ewert, J., and Selva, J.: Volcano observatory best practices (VOBP) workshops - a summary of findings and best-practice recommendations, Journal of Applied Volcanology, 8, 2, 10.1186/s13617-019-0082-8, 2019.

West, S. E., Bowyer, C. J., Apondo, W., Büker, P., Cinderby, S., Gray, C. M., Hahn, M., Lambe, F., Loh, M., Medcalf, A., Muhoza, C., Muindi, K., Njoora, T. K., Twigg, M. M., Waelde, C., Walnycki, A., Wainwright, M., Wendler, J., Wilson, M., and Price, H. D.: Using a co-created transdisciplinary approach to explore the complexity of air pollution in informal settlements, Humanities and Social Sciences Communications, 8, 285, 10.1057/s41599-021-00969-6, 2021.

Line 137. You provide examples of "outreach," but because this paper is using the term in a specific way and for analysis, it would be helpful to provide a specific definition of 'outreach.' For example, "In the paper we use the term "outreach" to represent interactions where individuals we classify as scientists do x, y z… (be explicit with what you classified as outreach and why).

**We agree that this distinction is required but believe the current wording provides this: '… outreach (e.g. to industry experts, the media, and decision-makers)'. Nonetheless, we now**

**identify this topic in our limitations section, noting that we had to be quite broad in determining the in-reach/outreach status since we made this classification based on Twitter profile information but without knowledge of a given users actual InSAR expertise. We suggest that self-reporting in feedback surveys would be a good way to determine the InSAR applications and level of user expertise.**

4.4 Limitations and other considerations
Although widely used for geoscience communications, InSAR communications over Twitter represent one social-media platform and other platforms and dissemination routes that are more relevant to specific stakeholders could be considered. Additionally, identifying types of InSAR users was limited to profile description classification in our study, producing scientist vs non-scientists groups that were used to infer whether communications were in-reach or outreach. However, these groupings are not exclusive and preclude a more comprehensive evaluation of how InSAR data are communicated and used by scientists, industry experts, the media, and decision-makers for example. Self-reporting in feedback surveys could help address this.

**Data and Methodology:**
I recommend adding a "roadmap" paragraph after section 2 and before 2.1 where you describe the various methods you used and what they were for. Then in the subsections you can expand on the finer details.
2.1.
**We have now added a roadmap paragraph as suggested:**
To investigate the prevalence and types of InSAR data communication over Twitter, we first collected and analysed a database of InSAR themed Tweets, which were also used to derive the community of InSAR Twitter users by analysing user connections and key words in profile descriptions. Engagement with open-access LiCSAR InSAR data was then evaluated using website analytics from the LiCSAR portal, combined with a user feedback survey. We then used the Earthquake InSAR Data Provider (EIDP) and the COMET Volcano Deformation Portal as example use cases of LiCSAR data to determine how these data align with those communicated using Twitter, and to identify directions to improve these InSAR data dissemination.

Line 145. I recommend adding a topic sentence to summarize the different analyses and data types pulled from Twitter before jumping into specifics of analysis packages.
**We have added the following text to start this section:**
Twitter was used to derive InSAR communications (tweets), including how widely the tweets were circulated (retweeted) through a network of users, each with profile descriptions that often indicate professions and interests.

Line 146. "Twitter data are increasingly used in scientific analysis." I don't think this is needed since data mining in Twitter are common practice these days.
**We have removed this sentence.**

Line 165. In your description of how you classify scientists and non-scientists, can you clarify

where government agencies that focus on science would be classified? I am guessing as "science," but government agencies are listed under the "non-scientist" category.

**Yes - If the twitter profile description indicated that a government agency was a scientific or research orientated agency for example, it would be listed under the scientific category. We have clarified the text as below:**

Examples of the 'scientist' class included: 'professor', 'phd', 'post doc', 'doctoral', and 'research scientist' and could be within academia (universities) or applied and research-focussed industry or government agencies.

2.2

2.2.1 – I like that you included ethical considerations for Twitter users, but perhaps this can go toward the end of the paper in a "limitations and other considerations" section?

**We have moved this section to the limitations and other considerations section in the discussion.**

Lines 1 189-190. *"Responses were also solicited by COMET members through peer networks and existing partnerships. COMET membership was asked as a survey question to distinguish these responses."*

Did you send COMET members the feedback survey directly? Were they asked only one survey question? Was this upon signing up for membership or for current members? Please be more explicit here with how you collected this information.

**We have modified the text below to state that we distributed a link to the full survey but this was ad hoc. It was primarily to ensure that partners we knew were using LiCSAR data had an opportunity to give feedback if they were not already aware of the form.**

Responses were also informally solicited by COMET members through peer networks and existing partnerships by sending a link to the feedback form.

2.4

The content in this section seems beyond the scope of this paper as it is not summarizing users of InSAR but rather gathering and providing recommendations as well as discussing a new tool that address those recommendations. With only 10 survey responses, I am not sure it is worth reporting the results here. These responses are valuable, but would carry more weight if there were more and solicited from both developing and more established countries to see similarities and differences in their data needs. Since this paper examines general use of InSAR, I am not sure if this narrowing of users to developing countries/overseas territories is appropriate in this paper. Additionally, authors go into a specific tool they developed with these recommendations which also seems to be outside the scope of this paper. I recommend removing this section.

**We acknowledge that this feedback was acquired and presented in a different way to the other systems discussed; however, we do believe it provides an important discussion topic to ensure portal development considers the end user requirements, and potential barriers to using satellite data. In this sense, it adds a discussion topic that contrasts with the more one-way flow of information disseminated by the Earthquake InSAR Data Provider for example. Since this information provides useful insight, and we do not believe it could form a**

**standalone paper, we have reworked and streamlined the volcano portal sections (2.4, 3.2.3, and 4.2.3) to consider this feedback.**

Line 216. This is getting into results from the 10 questionnaires and should not be in the methods section.
**This sentence was removed.**

Results:
3.1
Line 264. Missing comma between 2021 and respectively.
**Corrected.**

Figure 3. Recommend adjusting caption into to read "Timeline of InSAR tweets from September 2020 to February 2022…."
**Amended as suggested.**

Figure 4. Though interesting, I think word clouds a.- c. are unnecessary and can be removed. Bar graphs d and e are more helpful for understanding.
**We have removed the world clouds from this figure and moved them to the supplement.**

Figure 7. This figure is very busy and hard to see what the most important aspect is. It seems like 7c is most relevant for this paper. I am not sure a. and b. are helping readers interpret results. If they are interested in what the portal looks like, they can go to the website.
**corresponding references to Figure 7 in the text.**

Line 362. I am guessing the data frame is the region a user is pulling the data from. Can you please briefly describe what this is here for those who may be less familiar?
**Correct. We have now added the following text to explicitly refer to the frame IDs on the LiCSAR Portal.**
(individual frame IDs on the LICSAR portal)

Figure 8. Pie charts are best for responses with 3 or less categories. I recommend adapting this into bar charts similar to a-c for easier interpretation.
**We have adapted this figure as suggested.**

Figures 9 and 10. Are these full images needed? This paper has many figures and I wonder if they are all necessary for in-text reference? Could these be added as supplements or paired down to only include the most important pieces? As a reader, it is somewhat overwhelming to tease out important aspects of these multipart figures.
**We have now removed panel (a) from Figure 9 to leave the overview of the earthquake event page. We have left Figure 10 unmodified as we think it is important that the reader can see examples of interferograms with and without an earthquake signal.**

Figure 11 and Table 4: Similar to a previous comment. Could the text from Table 4 be added to the top of the panels in Figure 11 so readers could easily identify the event that ties to the page view count without having to flip between the table and figure? See example below:

[Figure]

**Thank you for this suggestion. We have made this change.**

3.2.3.
This section goes into detail of a specific portal and changes made from recommendations from a small sample of colleagues. Again, I think because this goes into such specifics it is beyond the scope of the paper outlined in the abstract and should be included in a separate manuscript.
**We have reworked the volcano portal sections in response to both reviewer comments to restructure how this feedback is integrated. We have also reduced the length of these sections to distil the key discussion points.**

**Discussion:**
4.1
Line 434 – 437. Nice topic sentence for this section.
In your research did you find that non-expert users were misinterpreting the data or information given the potential interpretation challenges for the SAR/inSAR data products? It would be nice to see some discussion of this.
**Whilst we agree that this would be a valuable topic of discussion, we did not analyse individual tweets in this detail, and it would potentially be contentious. However, we do refer to this topic in the existing text with reference to Figure S3. Here we show an atmospheric effect that produced a signal that looked like deformation at Agung Volcano.**

**'Whilst there is no guarantee that this network of users acts to positively filter InSAR data, for example, to mitigate common challenges such as misinterpretations of atmospheric effects as a deformation signal (for which we show a tweeted example of relating to Agung Volcano in Figure S3'.**

4.2
Line 491 – 512. This discussion may be outside of the scope of this paper or could be reduced in size as it gets a bit into the weeds.
**We have simplified this section and reworked as outlined above.**

4.3
Figure 12. I would recommend updating this figure. To make clear how the stages of disaster management relate to InSAR data product. Right now, they are sitting on the outside of the figure and it is not clear what their placement means or how it relates to the SAR information.

**We have updated this figure to move the disaster management headings into the figure as suggested. We have also added some visual icons relating to the key themes.**

4.4
Should this section name also include InSAR, "4.4 Future design and dissemination of LiCSAR **and InSAR** products" since the recommendations apply to both?
**We have made this change as suggested.**

Line 585-600. I would recommend bolding each recommendation so it stands apart from the description of the recommendation.
**We have amended this as suggested.**

**Conclusion:**
What is written in the conclusion summarizes part of what is discussed in the paper. Apart from analyzing Twitter, authors also looked at surveys, and other analytics from data platforms, and discussed a portal that was developed, as well as international feedback they received. If all of this is to remain in the paper, it should also be mentioned in the conclusion. However, my recommendation remains to refine the paper to fit what is currently in the conclusion and take out the additional pieces to develop a second paper.
I recommend including a limitations section for this paper. The Twitter ethics statement can also be included with this section.
**We have included mention of this additional feedback in the context of better integrating InSAR data dissemination into disaster risk reduction strategies.**
We presented an example of using volcano observatory user feedback to tailor the COMET Volcano Deformation Portal, by considering solutions to barriers to uptake of satellite data; however, in this case we drew on a small number (n=10) of respondents and continued feedback should be sought. We summarised our findings into five recommendations including both LiCSAR development and broader InSAR communications. For example, effectively communicating uncertainties, such as a spatially distributed displacement uncertainty, or uncertainties that accumulate through processing steps including atmospheric correction, are critical given the often complex interpretability of InSAR products.
**We also now include a 'Limitations and other considerations' section that includes the Twitter ethics information.**

---

## Author Comment (AC2)

**Response to Geoscience Communication Reviewer 2 (Ailsa Naismith)**

**Manuscript:** https://doi.org/10.5194/gc-2022-15
**Review**: https://doi.org/10.5194/gc-2022-15-RC2
**Response date:** 27-March-2023

Dear Alisa,
Thank you for your positive feedback and comments on our manuscript. We look forward to incorporating your suggestions in a revised manuscript submission. Please find our initial response to your specific comments in **bold**, with revised text in green.
Sincerely,
C. Scott Watson (on behalf of all co-authors)

**General comments:**

I think this is a very valuable subject for study, and I think that the authors have done the subject justice in this comprehensive and thoughtful manuscript. The work is coherent and well-reasoned throughout, with consideration of the multiple stakeholders that use InSAR data and specific areas where the authors show they have considered how this data may be processed and communicated ethically. The abstract is clear and covers the aim, results, and most of the important implications of the study. The manuscript is well-structured and easy to read throughout; the authors write fluently. Figures are easily understandable and clearly illustrate the relevant points in the manuscript, while the tables give a good summary of the meaning behind and application of different InSAR products. Well done to the designers, and to all involved in this study!

I have a couple of suggestions to make that I hope will further improve the quality of the manuscript. First, the abstract highlights the current challenge in communicating uncertainties with these data. I think this is an important and interesting challenge, but in the manuscript this section was shorter than I'd expected. It appears in the Conclusions, which references Figure 8c that specifies users strongly desire uncertainties included in future products. Do you have more information as to what uncertainties these might include? Adding 1-2 sentences of greater detail on uncertainties would bolster these conclusions – and given you end the abstract with them, they do feature quite prominently to a reader.

The second suggestion I would make would be to include a short section on the limitations of this study. I was surprised to not find any mention of the limitations of using Twitter in the first place. As the authors note, Twitter is used extensively by academic and industry scientists, and scientists in this study express that they want to communicate these data with non-scientists (i.e., outreach). But how much is Twitter used by these non-scientists? Exploring what platforms are used by these stakeholders should obviously not be added to this manuscript – it's another study entirely. Given that at least some of your colleagues (e.g., in Ethiopia) consider general education about the potential of satellite radar in volcanology to be important, I think there's scope to mention how Twitter is used only by some stakeholders (for instance, civil protection in the place I work in often

use Instagram and Facebook for communications). Including a section on study limitations could also include a brief consideration on misinformation related to communication of InSAR data. I am uncertain as to how big a problem it is, but certainly misinformation around other volcanic hazards shared on social media. If your results do not give any evidence to contribute to the discussion on misinformation around InSAR data, perhaps it would be worthwhile to signpost the possibility of misinformation so that it might be explored in a subsequent study.

My third suggestion is regarding Sections 2.4 (Data: COMET Volcano Deformation Portal) and 3.2.3 (Results: Engagement with the COMET Volcano Deformation Portal). These sections are fascinating and I think deserve more attention. However, I see a certain challenge in that the participants in each of these sections are relatively few (10 respondents in Section 2.4, and two responses in Section 3.2.3). Furthermore, they constitute a different kind of engagement to Twitter, in that responses and feedback are much more individual. I think the barriers to uptake of satellite data you present in Table 2 is very valuable, and I would like to see the engagement with this portal and future directions explored in more depth than what is presented in Section 3.2.3. With all that said, I recognize that this paper is already long at 31 pages, and that adding more detail to Sections 2.4 and 3.2.3 would make it even longer. Do you have enough data to present these sections as part of a separate, shorter second paper? That would be my recommendation. I think it would have the dual benefit of honing the focus of this excellent paper and giving sufficient space and respect to the interesting but too-short results presented in these two sections.

**In response to your general comments:**
**We have expanded our concluding point on uncertainties to provide some examples.**
Effectively communicating uncertainties, such as a spatially distributed displacement uncertainty, or uncertainties that accumulate through processing steps such as atmospheric correction, is also critical given the often complex interpretability of InSAR products.

**We have decided not to split this paper into two since we do not believe a second paper would include enough content. We also believe that including the Volcano Portal adds valuable insight into how such portals should be developed in response to user input. However, we recognise that both reviewers identified some of the Volcano Portal information as a divergence from the main theme of the paper and we have therefore reworked the volcano portal sections (2.4, 3.2.3, and 4.2.3) to integrate this important but small set of feedback. We have added a short limitations and other considerations section where we identify the issues in identifying stakeholders based on Twitter profile information alone. We have also included the ethical considerations surrounding the use of Twitter data here.**

**Specific comments**
COMET Volcano Deformation Portal and LiCSAR portal – AMAZING that these tools are out there, and that they are being used to engage with countries with active volcanoes worldwide!
Lines 125 – 128: these are very important and interesting questions. I read these as questions you might be answering in your study, and so was a little put out that they didn't appear again. Could you perhaps add a sentence to clarify that these are not the research questions you will be answering in this study but will instead be answered in another paper?

**We have clarified that these considerations provide an important context, but were not specifically evaluated in this study:**

We do not evaluate data use for specific disaster events in this study; however, these considerations provide an important context to the dissemination and discussion of data products for individuals or teams that are remote to the event, for example: (1) when and what data to release and with what level of interpretation, considering interpretation differences and underlying uncertainties; (2) considering whether releasing data will distract from official advice, or undermine local teams; (3) whether releasing the data can contribute to public safety, even if results are only preliminary.

Line 137: I think it's very interesting that you have distinguished in-reach from outreach, to see how much data moves outside the bubble. Great! Suggestion to add "… and the types of data that are communicated *in each case"* to emphasize that you will be studying the different data types for in-reach vs outreach.
**Amended as suggested.**

Line 146: on excluding retweets: I imagine this reduces the dataset to a manageable size, but do you then remove the possibility of exploring misinformation through images attributed to other events? I don't think you should add in data RTs, but I suggest that you could signpost the possibility of exploring retweets to identify a direction for future studies – that is, if misinformation is an issue for InSAR data communicating natural hazards (it certainly is for other types of data relating to volcanic hazards).
**We have added the following text to address this point:**

We excluded retweets to avoid considering the same tweet text multiple times, though we do consider the retweet count for each tweet. Retweets reveal how and what information is spread through Twitter, so could provide useful context on the spread of misinformation in relation to natural hazards.

Line 155: "(2) the tweeters of the top ten most retweeted tweets." Congratulations on this excellent tongue-twister.
**Unintentional but thank you.**

Lines 155 – 156: I find the wording of this pretty confusing. Can I clarify: *"top ten InSAR tweeters, (1) ranked by number of tweets, and (2) the tweeters of the top ten most retweeted tweets"* – I read this to mean that (1) are the ten accounts with the largest number of tweets in your study period, and (2) the accounts from which the ten tweets with most RTs come from during our study period. Is this correct? If not, I would suggest rewording to make this clearer.

This is correct.

Line 184: starting when in 2020 for Wordpress?
**We have added 'January'.**

Line 185: can you suggest a reason why the two are not directly compatible?
**Unfortunately, we could not determine how the two platforms were likely to differ, only that they potentially do (but may not). The sudden increase in page views from the end of 2019 to**

**2020 when we switched page tracking tools also coincided with redevelopment of the portal (hence the reason the tracking platform was changed), so it is difficult to know if the tracking tool influenced the change, or if the new portal was simply gaining more views. The increase 2020-2021 is obviously much clearer and unrelated to the swtich.**
**We have changed 'likely' to 'potentially'.**
however, the two analytics are potentially not directly comparable

Line 210: what language did you send the questionnaires in? Please specify in text.
**We have added 'in English'.**

Lines 214 – 215: the number of volcanoes … amazing! Was this Guatemala? I hear a different answer every time I ask.
**We have removed this line in our restructure and streamlining of the volcano portal sections.**

Table 2: I think this is an excellent table. Thoughts:
- Is there any significance to the different colours of the table columns? White looks like scientific problems, grey like capacity or resources. Please explain the difference in your table caption.
**Thank you. We have removed these cell colours to conform to the journal production guidelines.**

- This is perhaps my ignorance, but "automated processing" to me reads as building capacity for automated processing within institutions themselves. Wouldn't this require more computing capacity and high-speed internet, which you state is outwith the capacity of these institutions? Would "ready-processed and analysed data" be a more precise way of describing what you mean? (I lifted this from your text, which I think describes the challenges neatly.
**Agreed, we have changed as suggested.**

Lines 414 – 418: very, very interesting about the large difference between Ecuador and Ethiopia about the use of the portal – former in-reach (QC) and latter, outreach. I would love to read more about this, especially if and how you plan to continue seeking feedback from portal users and then implementing it. However, as I mention in my General Comments, this seems to be a different direction from the majority of the paper. Perhaps consider removing this section and sending to another one.
**We have removed this comparison in our restructure and streamlining of the volcano portal sections.**

Lines 502 – 504: "*We consider regular communicationl of volcano portal updates to be a better strategy than automatic direct social media responses to eruptions or unrest, due to the potential for eclipsing or distracting from the official communications from local volcano observatories and civil protection.*" – this is a very thoughtful and responsible stance for doing ethical science. Congrats!
**Thanks!**

Lines 514 – 527: *"Overall, InSAR Twitter was orientated towards natural hazards and specifically disaster or event response, rather than preparedness or disaster risk reduction, which reflects the current affairs nature of the Twitter platform (Petrovic et al., 2013; 515 Murthy, 2011; Acar and*

*Muraki, 2011)."* – this is very interesting, and as I wrote in the General Comments, I think it's important to consider that people involved in preparedness or DRR do not use Twitter as their primary platform. I think your following sentence: *"It would therefore be useful to investigate the links between data dissemination, knowledge building over Twitter, and local event response to determine the specific applications and benefits to disaster 525 management (Figure 12)"* is very valuable. Have you ever seen a study that does this? If so, I would recommend referencing it here. If not, I think you could include this as a Recommendation. Perhaps after your recommendations for LiCSAR development and InSAR communications, you could include this as a separate Recommendation for integrated communication of natural hazards? It would be a highly valuable but complicated and intricate study.

**We are not aware of such a study so have included a recommendation as suggested.**

Recommendation (5): Evaluate ways to better integrate natural hazard communications across multiple platforms. For example, linking data dissemination,  collaborative online knowledge building, and subsequent applications in local event response or disaster risk reduction strategies.

Technical Corrections
- Line 59: "in addition *or* anthropogenic signals" – should be "to", perhaps?
**Corrected to 'to'.**

- Line 74: "towards or *award* from the satellite" – should be "away", perhaps?
**Corrected to 'away'.**

- Line 126: "consideration for teams that remote to the event" – add "are" before "remote"
**Corrected to add 'are'.**

- Line 174: "*though* acceptance of Twitter's user terms" – should be "through"
**Corrected.**

- Line 175: "Tweets" – lower-case
**Corrected.**

- Lines 246 – 248: you're missing a bracket to close the "(this …"
**Corrected.**

- Line 359: remove comma after "interferograms"
**Corrected.**

- Line 440: remove comma after "particularly"
**Corrected.**

- Line 536: "offer *one* mechanism" – this is ambiguous – what is that mechanism? Or do you mean the threads are the mechanism? In which case, I'd switch the "one" to "a".
**Corrected to 'a'.**

---

## Author Response (AR1)

**Response to Editor**
**Manuscript:** https://doi.org/10.5194/gc-2022-15
**Response date:** 24-April-2023

Dear Scott and coauthors,

Thank you for engaging so thoroughly in the review process. Based on the reviewers' comments and on your responses I suggest minor revisions. Please revise your manuscript based on your responses to the reviewers.

A few additional comments:

While I recognize that splitting your manuscript in two is not possible due to lack of enough content, I agree with both reviewers that the volcano portal section and the results based on the small subset of feedback (10 responses and only from developing nations) is not a good fit for this manuscript. Please consider removing it from the main text.

I also suggest combining table 1 and figure 2 (as suggested by one of the reviewers) as it will improve its readability. For the revised figure, you may consider choosing, coherence, warped and unwrapped versions of the same interferogram as opposed to using different examples to drive your points and eliminate unnecessary details from the images you used in figure 2.

And finally, the work of Gill et al. (2021) on recommended actions for integration of natural hazard data into DRR work might be a useful read to include in your section 4 or 5: https://nhess.copernicus.org/articles/21/187/2021/

I look forward to reviewing your revised manuscript.

All the best,
Solmaz Mohadjer

Dear Solmaz Mohadjer,

Thank you for your additional comments on our manuscript.

- Following the reviews and after discussion with coauthors, we decided that we strongly wanted to retain the volcano portal sections in the text, so we substantially reworked and streamlined these sections and believed it was now much better aligned with the rest of the manuscript. However, we understand the issue with the small amount of feedback gathered so far. We believe that the volcano portal is a valuable communication tool and therefore discussion point

in the paper that complements the other portals discussed, would be of interest to the article readers, and is relevant to the high Twitter activity we observed during volcanic events. With this in mind, we have retained the volcano sections at reduced length and moved the corresponding Table 2 (*Barriers to uptake of satellite data identified by volcano observatories. Information is summarised from the responses from a range of official development assistance countries to a 2017 survey*) from the main text to the supplement. We have also moved the description of the survey from section 2.4 to the supplement.

- We have revised Figure 2 to combine with the previous Table 1 as suggested by the reviewer and yourself. This now forms a landscape figure.
- Thank you for highlighting the work of Gill et al. (2021), which we now refer to in Sections 4.2.3 and 4.3.

We have uploaded the revised manuscript with tracked changes.

Sincerely,
C. Scott Watson (on behalf of all co-authors)